



# The Kenya Rift revisited: insights into lithospheric strength through data-driven 3D gravity and thermal modelling

Judith Sippel[1], Christian Meeßen[1,2], Mauro Cacace[1], James Mechie[1], Stewart Fishwick[3], Christian Heine[4], Magdalena Scheck-Wenderoth[1], Manfred R. Strecker[2]

[1]GFZ German Research Centre for Geosciences, Sections 6.1 & 2.2, Telegrafenberg, 14473 Potsdam, Germany
[2]Institute of Earth and Environmental Science, University of Potsdam, Potsdam, 14476, Germany
[3]Department of Geology, University of Leicester, Leicester,  LE1 7RH, UK
[4]New Ventures, Upstream International, Shell International Exploration & Production B.V., 2596 HR, The Hague, Netherlands

*Correspondence to*: Judith Sippel (sippel@gfz-potsdam.de)

## Abstract

We present 3D models that describe the present-day thermal and rheological state of the lithosphere of the greater Kenya Rift region aiming at a better understanding of the rift evolution, with a particular focus on plume-lithosphere interactions. The key methodology applied is the 3D integration of diverse geological and geophysical observations using gravity modelling. Accordingly, the resulting lithospheric-scale 3D density model is consistent with (i) reviewed descriptions of lithological variations in the sedimentary and volcanic cover, (ii) known trends in crust and mantle seismic velocities as revealed by seismic and seismological data, and (iii) the observed gravity field. This data-based model is the first to image a 3D density configuration of the crystalline crust for the entire region of Kenya and northern Tanzania. An Upper and a Basal Crustal Layer are differentiated, each composed of several domains of different average densities. We interpret these domains to trace back to the Precambrian terrane amalgamation associated with the East African Orogen and to magmatic processes during Mesozoic and Cenozoic rifting phases. In combination with seismic velocities, the densities of these crustal domains are indicative of compositional differences. The derived lithological trends have been used to parameterize steady-state thermal and rheological models. These models indicate that crustal and mantle temperatures decrease from the Kenya Rift in the west to eastern Kenya, while the integrated strength of the lithosphere increases. Thereby, the detailed strength configuration appears strongly controlled by the complex inherited crustal structure, which may have been decisive for the onset, localisation, and propagation of rifting.

## Keywords

Plume-lithosphere interactions, 3D crustal density, deep crustal composition, rift localisation



# 1 Introduction

Continental rifting involves lithospheric stresses imparted by thermally driven mantle upwelling (as featured by the active rift model) or far-field stresses generated by plate-boundary forces (as highlighted by the passive rift model; e.g. Turcotte and Emerman, 1983). Beside these extrinsic factors, localised stretching of lithospheric plates is controlled by the rheology

of rocks and thus by intrinsic factors such as the composition as well as the pressure and temperature configuration of the lithosphere (as demonstrated by forward numerical experiments, e.g. Watts and Burov, 2003, Huismans et al., 2005). Hence, to improve our understanding of how deformation localises in continental rifts, site-specific lithological heterogeneities have to be taken into account. To assess the compositional configuration of rifted lithosphere is challenging given that the continental crust is generally the product of a complex structural, magmatic and metamorphic history, leading to pronounced

anisotropies prone to guide extensional deformation processes. This is an important problem in the East African Rift System (EARS; Fig. 1a) and particularly the Kenya Rift, because this region has a long history of continental collision, subsequent orogen-parallel shearing and extensional faulting prior to the formation of the Cenozoic Kenya Rift (Burke, 1996). The key to a holistic lithological and physical description of the lithosphere lies in the 3D integration of different geological and geophysical observations. Here, we present data and lithology-driven numerical 3D models describing the present-day

thermal and rheological state of the lithosphere for the greater Kenya Rift region (Fig. 1a) to uncover major strength anomalies that are prone to localise deformation.

The Kenya Rift is regarded as the classical example of an active rift, whose initiation and protracted evolution have been fundamentally controlled by mantle dynamics until the present day. A continental-scale plate kinematic model (i.e., Stamps et al., 2008) reveals that East Africa is dominated by extensional processes that determine the separation of the Somalia and

Nubia plates with an approximate rate of 4.7 mm/yr determined for the Ethiopia-Kenya border area. Seismological studies in East Africa have imaged an upper-mantle low-velocity zone below the EARS that is commonly interpreted as a high-temperature anomaly (e.g. Mulibo and Nyblade, 2013), inferred to be connected to the lower-mantle sectors of the South African Superplume (e.g. Bagley and Nyblade, 2013). During the past 35-45 Ma the African tectonic plate has been moving northward relative to the East African plume as underscored by the volcanic and topographic evolution of the EARS (e.g.

Ebinger and Sleep, 1998; Moucha and Forte, 2011; Wichura et al., 2015). Subsequent to regional doming, but possibly also during updoming, extensional tectonics and volcanism have affected the region from the Turkana area in northern Kenya to the Tanzania Divergence (Fig. 1a; e.g., Morley et al., 1999). There is consensus that crustal extension and magmatism in the Kenya Rift are related to plume-lithosphere interactions (e.g. Mechie et al., 1997). Halldórsson et al. (2014) suggested that regionally overlapping amagmatic and magmatic sectors of rift initiation in East Africa follow the asymmetric impingement

of a single mantle plume at the base of the lithosphere in the transition between the Tanzania Craton and areas to the east. However, detailed regional-scale assessments on the interaction between mantle dynamics and the overlying compositionally and structurally heterogeneous lithospheric plates of the region have not been attempted to ultimately explain rift localisation in Kenya and the mechanical predisposition of fault propagation. To improve our understanding of where and how the

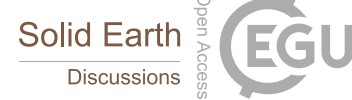

extensional deformation patterns have evolved, we develop a lithospheric-scale 3D structural model covering the Kenya Rift region (Fig. 1a; black rectangle) that provides insights into the Cenozoic rheological configuration.

The formation of the crust in East and Southeast Africa dates back to the Neoproterozoic and early Paleozoic. Two tectonothermal phases known as the East African Orogeny (at ≈650-620 Ma) and the Kuunga Orogeny (at ≈600-500 Ma) led

to the amalgamation of numerous terranes to form central Gondwana (e.g., Fritz et al., 2013). The East African Orogeny has resulted from collisions of the Arabian Nubian Shield and its southward continuation, the Mozambique Belt (Holmes, 1951), with the Tanzania (Nyanzian) Craton to the west and the Azania microcontinent to the east; this orogeny is responsible for the juxtaposition of compositionally different crustal blocks in the study area (Fig. 1b). Many authors have put forward structural and compositional differences between the Tanzania Craton and the weaker Mozambique Belt as controlling

factors for the localisation of the Cenozoic Kenya Rift (e.g. McConnell, 1972; Smith and Mosley, 1993; Tesha et al., 1997). Alternatively, Koptev et al. (2015) present a thermo-mechanical numerical model that simulates how (under tensional far-field stress) the proposed mantle plume is deflected by the lithospheric keel of the Tanzania Craton to cause the initiation of a rift system east of it. In this rheologically consistent model, strain localisation is due to lithospheric thinning and channelised flow of the plume material.

Our approach focuses on the structure and rheological configuration of the lithosphere as a key to improve our understanding of the Kenya Rift tectonic evolution. In general, predictions on the strength of the lithosphere require knowledge about its compositional and thermal configuration (e.g. Goetze and Evans, 1979; Burov, 2011). For the study area it is particularly challenging to assess how the complex basement geology continues into the deeper crust. The results of the Kenya Rift International Seismic Project (KRISP, 1985-1994) provide important constraints on the nature of the crust along five

regional sections extending along and across the rift (Fig. 1b). We have integrated the processed and interpreted KRISP refraction seismic profiles (Khan et al., 1999) with various other geological and geophysical observations (from sedimentary fills, crustal and mantle characteristics) to perform 3D gravity modelling and develop a lithospheric scale 3D density model. In this 3D data integration process, P- and S-wave velocity models of the mantle (e.g. Achauer and Masson, 2002) play an important role, especially as mantle anomalies affect the regional gravity field significantly (e.g. Achauer, 1992; Ravat et al.,

1999; Mariita and Keller, 2007). The spatially continuous gravity signal facilitates a 3D investigation of the crust and thus guided extrapolations of crustal properties beyond the KRISP profiles. Finally, we have derived lithological variations within the crust and discussed them with respect to their potential origin. The interdisciplinary integration of data thus allows us to assign thermal and rheological properties to lithological model units to assess the present-day thermal and yield strength configuration of the lithosphere. The resulting 3D rheological models ultimately reflect the interplay between the observed

mantle thermal anomaly and compositionally different crustal domains which we discuss with respect to rift localisation and propagation, seismicity and volcanism.





## 2 Modelling approach: 3D data integration and gravity modelling

The key methodology of this study is 3D gravity modelling. This involves determining a 3D density configuration of the subsurface, for which the calculated gravity response reproduces the measured gravity field. As the potential field modelling techniques are inherently non-unique, our goal has been to minimise the number of free parameters in advance. We have

followed a strategy to (1) take into account available various geological and geophysical data to constrain a starting density model with defined interfaces and densities for the sedimentary and volcanic cover, the crystalline crust, and the mantle; and (2) modify the density configuration of the crust within the data constraints to fit the observed gravity. We have used the 3D potential field modelling software IGMAS+ (©Transinsight GmbH) which allows interactively changing 3D density configurations while simultaneously maintaining visual control over the calculated gravity response (Schmidt et al., 2011).

**2.1 Constraints on the density configuration of the sedimentary and volcanic rocks**

Information on the depth of the base of sedimentary and volcanic rocks (Fig. 2a) is combined from two sources in the study area: within the political boundaries of Kenya and offshore, the Geological Map of Kenya provides contour lines for the depth to the crystalline basement (spaced at 1 km depth intervals; Beicip, 1987). For the modelled areas outside of Kenya, basement depth constraints have been derived from a global map of total sediment thickness (Fig. 2a). This sediment

thickness estimate is based on isopachs derived from the Exxon Tectonic Map of the World (Exxon, 1985). For the global model, the digitised isopachs were gridded using a spherical splines-in-tension algorithm (Wessel et al., 2013) onto a 6 arc-minute raster, taking into account outcropping basement rocks as determined by the USGS World Energy Project regional geological map data (USGS, 2012).

The top of the basement (Fig. 2a) is overlain by sedimentary and volcanic rocks of Permo-Carboniferous to Holocene ages

(Beicip, 1987) and its geometry reflects different phases of sedimentary basin formation and localised subsidence. For instance, the Mandera and Lamu Basins in eastern Kenya (Fig. 2a) are regarded as the north-easternmost extension of the Karoo rift system, the initiation of which was related to the Late Carboniferous-Early Permian assembly and subsequent breakup of Pangea (Catuneanu et al., 2005). After an early phase of eastward rifting of Madagascar (and India) away from the conjugate block of Kenya and northern Tanzania (e.g. Reeves et al., 2002), from ≈185-180 Ma Madagascar moved

southwards (e.g. Cox, 1992), leading to the formation of oceanic crust in the Indian Ocean (at <166-152 Ma; Seton et al., 2012) and transforming the Lamu Basin area into a passive margin setting.

Farther west, the oldest structural elements of the NW-SE oriented Anza Basin (Fig. 2a; Bosworth and Morley, 1994) and the N S oriented Lotikipi Plain, Turkana, Lokichar, and North Kerio basins (e.g., Morley, 1999) began forming during the Cretaceous and continued into the Cenozoic (Foster and Gleadow, 1996; Morley, 1999; Tiercelin et al., 2012). The Anza

Basin has been regarded as part of the E-W striking Central African Rift System (Guiraud et al., 2005; Heine et al., 2013), which formed under the influence of (i) the northeastward movement of the Arabian-Nubian block, (ii) ongoing seafloor



spreading between Madagascar and East Africa (which ceased at around 120 Ma; Seton et al., 2012), and (iii) the opening of the South Atlantic (since 132 Ma).

The Cenozoic-aged, broadly N-S trending Kenya Rift comprises the northern, central and southern segments. In the northern Kenya Rift, earliest extension earliest extension began during the Paleo-Eocene (Morley et al., 1992; Ebinger and Scholz, 2012). New thermo-chronological data from the Elgeyo Escarpment in the northern sector of the central rift segment also reveal Paleo-Eocene rift initiation, subsequent subsidence and heating, followed by renewed cooling and formation of major rift-bounding faults after 15 Ma (Torres Acosta et al., 2015). Along the Nguruman Escarpment of the southern Kenya rift extensional faulting is shown to have started at approximately 7 Ma (Crossley, 1979). In contrast, farther south within the Tanzania Divergence, thermo-chronological data suggests that extensional faulting and cooling began during the Cretaceous and continued into the Paleo-Eocene (Noble et al., 1997; Mbede, 2001). Rifting is generally thought to have followed shortly after volcanism started in the different rift segments (e.g. Morley et al., 1992). The oldest volcanics in northernmost Kenya are as old as ≈39-45 Ma (e.g. Ebinger et al., 2000), while volcanism reached the intersection between the northern/central Kenya Rift and the Nyanza Rift (Fig. 2a) at ≈20 Ma (Pickford, 1982; Fitch et al., 1985) and the oldest volcanics in the southern Kenya Rift are between 20 and 16 Ma (Baker et al., 1972; Chapman et al., 1978; Smith, 1994; Hay et al., 1995). A recent synopsis on the onset of volcanic activity in East Africa by Michon (2015) suggests that earlier interpretations of a N-S migration of volcanism and tectonic activity (Nyblade and Brazier, 2002) may not apply to the Kenya Rift and that these processes were rather highly disparate in space and time in the EARS as suggested previously by Zeyen et al. (1997).

The difference in elevation between the topography/bathymetry (Fig. 1a) and the base of the sedimentary and volcanic rocks (Fig. 2a) defines the thickness of the rift-basin fill (Fig. 2b). The largest cumulative thicknesses of locally >10 km are found in the Mandera, Lamu and Anza basins in eastern Kenya, while in the Kenya Rift deposits may locally attain thicknesses of up to 8 km (Mugisha et al., 1997; Hautot et al., 2000).

In general, the density of sedimentary and volcanic rocks depends on their mineralogical composition and porosity, the latter in turn being related to the local degree of compaction. According to differences in the prevailing lithological compositions as described in numerous studies (Table 1; Appendix A), we have subdivided the sedimentary and volcanic cover of the study area into seven domains of distinct density configurations (Fig. 2b; Table 2). The Mandera, Lamu and Anza Basins, for example, have been grouped into the Eastern Basins domain. Some of these domains are additionally separated into vertical sequences of sub-units with different densities, thus reflecting further lithological and/or porosity variations (Table 2; Appendix A).

## 2.2 Constraints on the density configuration of the crystalline crust

According to Fritz et al. (2013), five major tectonothermal domains of different ages and lithologies are juxtaposed against each other in the study area (Fig. 1b). From W to E, these are (i) the Nyanzian System (Clifford, 1970) of the Archean Tanzania Craton; (ii) the Western Granulites representing reworked pre-Neoproterozoic crust of the Mozambique Belt (Maboko, 1995; Möller et al., 1998);, (iii) the Eastern Granulites representing Neoproterozoic juvenile crust of the



Mozambique Belt (Möller et al., 1998; Maboko and Nakamura, 2002; Tenczer et al., 2006); (iv) the Neoproterozoic Arabian Nubian Shield; and (v) the microcontinent Azania representing reworked pre-Neoproterozoic crust (Fritz et al., 2013). Near-surface observations indicate that rocks of the Mozambique Belt and Arabian Nubian Shield structurally overlie the craton in the west and the microcontinent in the east (Fritz et al., 2013).

Table 3 provides an overview of the lithological variation across the five tectonothermal domains. The Nyanzian System of the Archean Tanzania Craton is a typical low-grade metamorphic greenstone belt assembly of metamorphosed volcanics, sediments and granites (e.g. Clifford, 1970). Since the Western Granulites mainly consist of low-grade metamorphic rocks of sedimentary and magmatic origin (Mosley, 1993), the name "granulites" is misleading although widely established (Fritz et al., 2005, 2013; Cutten et al., 2006). The Eastern Granulites largely consist of a basal unit of meta-igneous rocks (Tenczer et

al., 2006) and an upper unit of meta-sedimentary sequences (Fritz et al., 2005, 2009). According to Mosley (1993), the lithostratigraphic groups of the Eastern Granulites and the Arabian Nubian Shield are very similar. The metamorphic volcano-sedimentary sequences of the Arabian Nubian Shield, however, largely belong to several arc-magmatic terranes and include numerous ophiolites (Fritz et al., 2013).

Due to the thick cover of Mesozoic sedimentary rocks in the eastern parts of the study area (Fig. 2b), the crustal composition

of Azania has largely been inferred from other parts of this microcontinent exposed in Madagascar (e.g. Randriamamonjy, 2006) and Somalia. Accordingly, the crust of Azania mainly consists of orthogneisses (Collins and Pisarevsky, 2005 and references therein) that are described as granites north of the Anza Basin (Mosley, 1993) or granite gneisses and granites overlain by meta-sedimentary sequences in western Madagascar (Collins and Pisarevsky, 2005; Randriamamonjy, 2006).

*Geophysical constraints*

For the initial density model, we generated a Moho-depth configuration by interpolation between data derived from diverse sources and scattered across the study area (Fig. 3a): most importantly the Moho derived from the KRISP refraction seismic profiles (e.g. Khan et al., 1999) and receiver function data (Tugume et al., 2012, 2013); but also crust-mantle boundaries as imaged by a regional, seismically and gravity-constrained 3D density model (Woldetinsae, 2005) as well as by the global

model "Litho 1.0" (Pasyanos et al., 2014). The obtained Moho reveals two major highs with depths of <25 km (Fig. 3a): one extending N-S underneath the surface expression of the Turkana Basin domain (cf. Fig. 2a) and a second one delineating the the oceanic crustal domain of the Indian Ocean. The largest Moho depths of >40 km are observed in the northwestern parts of the study area, and locally on the eastern flanks of the rift (KRISP line F; Fig. 3a).

The difference in depth between the top of the basement (Fig. 2a) and the Moho (Fig. 3a) defines the thickness of the

crystalline crust underlying the sedimentary and volcanic deposits (Fig. 3b). In the oceanic domain, crustal thicknesses are as low as 5-12 km, while for 60% of the continental domain crustal thicknesses are larger than 35 km. The Turkana and Northern Rift domains have thicknesses of the crystalline crust reduced to less than 16 km. Thinned crystalline crust also characterises the Eastern Basins domain, such as along the Anza Basin axis (with thicknesses of around 30 km; cf. Fig. 3b, Fig. 2a) and the Lamu Basin, close to the continent-ocean boundary, where values decrease to <15 km.



The KRISP refraction seismic profiles further provide information on intracrustal discontinuities (e.g. Khan et al., 1999; Fig. 4a). For each profile, seismic velocity information is available at grid points laterally spaced at 10 km intervals. Vertically, from the top of the basement down to the Moho, velocity is defined at 3-8 depth levels, depending on the XY-position and the complexity of the velocity models. According to these seismic profiles, the crust is widely structured into

three sub-horizontal layers, such as on lines ABC, E, and F (Khan et al., 1999). However, locally a four-layered crust (line G) or smaller-scale velocity discontinuities within the shallower crust do occur as well (line D; Fig. 4a).

A structural element that can be traced continuously over all profiles inside and outside the rift is what previous authors have referred to as the Basal Crustal Layer (Fig. 4a). While the Moho is depicted by a sudden increase in P-wave velocities to values of $v_p$>7.5 km s$^{-1}$, the top of the Basal Crustal Layer is identified as the depth at which velocities rise to values of

$v_p$≥6.7 km s$^{-1}$ (up to $v_p$≤7.1 km s$^{-1}$; Mechie et al., 1997; Khan et al., 1999). In contrast, crustal domains located between the Basal Crustal Layer and the sedimentary cover show an overall velocity range of $v_p$=5.9-6.65 km s$^{-1}$ and are, in the following sections, collectively referred to as the Upper Crustal Layer (locally comprised of upper, middle and lower crust).

Together with the top of the basement (Fig. 2a), the depth configurations of these velocity contrasts outline the thicknesses of the Upper Crustal Layer (Fig. 4b) and the Basal Crustal Layer (Fig. 4c). The Upper Crustal Layer thins from the rift flanks toward the rift; although less distinctly expressed, this trend is also observed for the Basal Crustal Layer (lines ABC, D, E,

G). Furthermore, both crustal layers continuously thin from S to N along the rift (line ABC). Finally, while the Upper Crustal Layer significantly thins towards the continent-ocean boundary (COB), the Basal Crustal Layer reveals its largest thicknesses in the SE parts of the study area.

To uncover lateral trends from the complex velocity structure of the Upper Crustal Layer, we have vertically averaged over

the observed interval velocities to obtain a mean velocity for each XY-position along the KRISP profiles. The resulting average velocities show an overall range of $v_p$=6.10-6.46 km s$^{-1}$ (Fig. 4d). In western Kenya (west of the stippled line in Fig. 4d), velocities of $v_{p,c}$<6.35 km s$^{-1}$ are abundant, while in eastern Kenya larger velocities of $v_{p,c}$≥6.40 km s$^{-1}$ prevail. Representative means for western and eastern Kenya would be $v_{p,c}$=6.325 km s$^{-1}$ and $v_{p,c}$=6.425 km s$^{-1}$, respectively. The relationship between crustal density ($\rho_c$ [kg m$^{-3}$]) and velocity ($v_{p,c}$ [km s$^{-1}$]) reading as

25         $$\rho_c = 378.8 * v_{p,c} + 350 \quad \text{(Eq. 1),}$$

is a modification of Birch's (1961) law (Appendix B). According to Equation 1, the mean velocities of the Upper Crustal Layer in western and eastern Kenya translate into densities of $\rho_c$≈2750 kg m$^{-3}$ and $\rho_c$≈2780 kg m$^{-3}$, respectively. One goal of performing the 3D gravity modelling is to test whether this general W-E velocity increase effectively corresponds with such a density increase in the Upper Crustal Layer.

We have applied the same density conversion function, Eq. (1), to the velocities of the Basal Crustal Layer (Fig. 4e). Accordingly, the deepest crust shows the largest velocities and densities ($\rho_c$≥3000 kg m$^{-3}$) in the S (lines F, G) and the smallest values ($\rho_c$<3000 kg m$^{-3}$) along the rift and farther to the NE (lines ABC, E).



### 2.3 Constraints on the density configuration of the mantle

To assess the 3D density configuration of the mantle, in particular the geometry of the low-velocity anomaly (LVA) underneath the rift, we have analysed published models of seismic P- and S-wave velocities.

#### 2.3.1 Analysis of P-wave velocity data

Ravat et al. (1999) have proposed a linear P-wave velocity-density relationship for the mantle underlying the Kenya Rift and its shoulders:

$$\rho_m = 2855 + 50 * v_{p,m} \quad \text{(Eq. 2)},$$

with $\rho_m$ [kg m$^{-3}$] being the mantle density and $v_{p,m}$ [km s$^{-1}$] its P-wave velocity. To assess this formulation (that is valid for depths down to ≈200 km), the authors have integrated diverse data from southern Kenya, such as (i) crustal velocities from

the KRISP refraction line D (Fig. 4a; Braile et al., 1994; Maguire et al., 1994), (ii) upper mantle velocities from the KRISP '85 teleseismic experiment (Slack et al., 1994; Fig. 5a) and (iii) gravity data (Maguire et al., 1994).

We have made use of this relationship to convert mantle P-wave velocity data to density. The KRISP refraction seismic profiles image mantle P-wave velocity from the Moho down to a maximum depth of 70 km (e.g. Khan et al., 1999). After performing a point-by-point velocity-to-density conversion of the data using Eq. (2), we have vertically averaged over the

resulting mantle densities for each XY-location. In this way, the gravity-relevant lateral density contrasts across the region were revealed (Fig. 5a).

One main finding of the KRISP survey is that mantle domains underlying the rift proper (e.g. KRISP line ABC) are characterised by lower velocities than mantle domains outside the rift (off-rift parts of KRISP lines D, E, F, G; e.g. Mechie et al., 1994). Accordingly, KRISP line D shows average mantle densities of $\rho_m$≈3270 kg m$^{-3}$ west and east of the rift, while

densities are as low as $\rho_m$≈3250 kg m$^{-3}$ right beneath the surface expression of the rift (Fig. 4a). This across-rift density difference of $\Delta\rho_m$≈20 kg m$^{-3}$ derives from a velocity difference of about $\Delta v_p$≈0.4 km s$^{-1}$ and is also observed along the across-rift line G further south (Fig. 5a). Along the strike of the rift, i.e. along KRISP line ABC, the mantle shows densities increasing from about 3240 kg m$^{-3}$ in the south to 3250 kg m$^{-3}$ in the north. A similar S to N increase in mantle density (≈15-20 kg m$^{-3}$) is observed outside the rift as revealed by the density difference between (i) KRISP lines D and E in the

north compared to (ii) lines F and G further in the south.

Besides the KRISP data, we have also analysed the teleseismic data of Slack et al. (1994) and Achauer and Masson (2002), both covering depths greater than 150 km. For the analysis of these teleseismic models, images from publications of the tomographic models have been georeferenced and the resulting scattered velocities converted to densities using Eq. (2). The tomographic P-wave velocity model presented by Achauer and Masson (2002; Fig. 5a) reveals a pronounced LVA under the

rift, with amplitudes as large as -10% relative to the Preliminary Reference Earth Model (PREM; Dziewonski and Anderson, 1981). Slack et al. (1994) described a more gradual change from a 12 %-velocity anomaly confined to the rift to a 6 %-velocity anomaly below its flanks. In both cases, the velocity perturbations delineate a LVA that is essentially confined to



the surface expression of the rift down to about 100 km depth. With increasing depths down to 300 km, the LVA widens while becoming more diffuse so that it cannot be delineated from the surrounding unperturbed mantle anymore in the tomography model of Achauer and Masson (2002).

According to Eq. (2), the P-wave velocity perturbation with respect to PREM as revealed by the model of Achauer and
Masson (2002) records an across-rift density variation of ≈7 kg m$^{-3}$ (3251-3258 kg m$^{-3}$) at depths of 35-70 km and ≈4 kg m$^{-3}$ (3251-3255 kg m$^{-3}$) at depths of 70-110 km. Thus, the density contrast of the mantle anomaly indicated by the tomography study is slightly smaller than the contrast revealed by the KRISP refraction seismic profiles (≈20-25 kg m$^{-3}$). However, the two datasets are consistent in terms of the location and spatial extent of the mantle-density anomaly.

Overall, P-wave velocities are indicative of a LVA that (i) is essentially confined to the surface expression of the rift, i.e.
only slightly widening down to depths of about 100 km and (ii) differs in density from its surroundings by ≤25 kg m$^{-3}$. Considering these observations we have constructed a starting density model of the shallow mantle (between the Moho and 100 km b.s.l.) that differentiates six density domains (Fig. 5a). The LVA is represented by a southern domain of 3240 kg m$^{-3}$ and a northern domain of 3250 kg m$^{-3}$. Thereby, the modelled western and eastern boundaries of the LVA are inclined (Fig. 5a) reflecting the observed downward widening of the LVA, which is more pronounced in the north (e.g. Simiyu and
Keller, 1997) than in the south (Slack et al., 1994; Achauer and Masson, 2002). Outside the rift, this starting density model differentiates four domains for the shallow mantle with densities that are larger by 15-25 kg m$^{-3}$ compared to the two rift-mantle domains (Fig. 5a). These four off-rift mantle regions also reveal a general increase in density from the south to the north.

### 2.3.2 Analysis of S-wave velocity data

Adams et al. (2012) present a quasi 3D S-wave velocity model of the upper mantle that is inverted from Rayleigh wave phase velocities as derived from teleseismic recordings on broadband stations located in Uganda, Tanzania and Kenya. The derived S-wave velocity model provides information on the mantle configuration for depths of 50-400 km and covers the Tanzania Craton and its adjacent western and eastern branches of the EARS, thereby also extending into the southwestern parts of Kenya. For a 3D analysis of this particular region, we have digitised and georeferenced four depth slices (at 100,
150, 200 and 250 km depth) and six profiles presented in the paper of Adams et al. (2012).

Since the model of Adams et al. (2012) only covers the SW parts of the study area, we have also utilised the results of an updated surface-wave tomography study based on Fishwick (2010). This study is likewise based on Rayleigh wave velocities, but using source-to-receiver paths that cover the entire African continent and surrounding oceans. Accordingly, the nodal points of the model of Fishwick (2010) are set at 1.5° intervals, i.e. at a larger distance compared to the phase
velocity inversion of Adams et al. (2012) performed for a grid with 0.5° node spacing. Vertically, Fishwick's (2010) tomographic models of S-wave velocity are defined at 25 km-depth intervals from 50 to 350 km depth.

Despite the differences between the two surface-wave tomography models in terms of utilised data and inversion procedures, they are consistent with respect to the main trends in absolute S-wave velocities across the SW parts of Kenya. Thus, we



have merged the two S-wave velocity models for subsequent modelling steps. In more detail, we have complemented the model of Adams et al. (2012) towards the N and E by the model of Fishwick (2010) and 3D interpolated the scattered point information to obtain a voxel grid of regular spacing of 50 km horizontally and 20 km vertically.

At a depth of 150 km (Fig. 5b), the combined model shows the lowest velocities of around 4.3 km s$^{-1}$ below the Kenya Rift,

increasing to 4.4 km s$^{-1}$ in the Eastern Basins domain and to >4.5 km s$^{-1}$ in the Tanzania Craton domain. Furthermore, the LVA underneath the Kenya Rift widens significantly from S to N. The difference between the LVA and higher velocities in the surroundings decreases with depth: at 100 km depth, the lateral variability amounts to 0.5 km s$^{-1}$, while at 200 km depth S-wave velocities differ maximally by 0.2 km s$^{-1}$. Below 200 km depth, the LVA widens significantly transforming into a continuous low-velocity layer that extends from the Kenya Rift across the Tanzania Craton to the western branch of the

EARS (e.g. Adams et al., 2012). Consequently, there are no gravity-relevant, lateral density contrasts to be expected from depths of >200 km.

To convert the observed mantle S-wave velocities into densities we have followed a two-stage approach. In a first step, we used the set of empirical equations and constants proposed by Priestley and McKenzie (2006) to convert mantle S-wave velocity to temperature. This non-linear relationship is assumed to be valid for any mantle composition, while being most

accurate for temperatures that exceed 1100°C and depths of >100 km. In a second step (Appendix C), we used a mantle composition proposed for the region (Mechie et al., 1994) and converted the calculated mantle temperatures to densities. Because of (i) the depth restriction of the velocity-density conversion, (ii) the good coverage of P-wave velocity data down to 100 km, and (iii) the very small variability in S-wave velocities at depths of >200 km, we have performed these conversions for each point of the S-wave velocity voxel grid between depths of 100-200 km.

Figures 5b-d show depth slices at 150 km b.s.l. extracted from the calculated 3D grids. According to the low S-wave velocities underneath the Kenya Rift and in the northern parts of the study area (Fig. 5b), the mantle shows the highest temperatures (Fig. 5c) and lowest densities (Fig. 5d) there. This high-temperature, low-density anomaly widens from the Kenya Rift towards the north (i.e. the Ethiopian and Afar Rift systems). East of the Kenya Rift, density increases to moderate values under the Eastern Basins domain while decreasing again towards the Indian oceanic domain in the southeast. The

largest densities are found in the Tanzania Craton domain (Fig. 5d; cf. Fig. 1b). Overall, the absolute densities at a depth of 150 km scatter around a mean of 3324 kg m$^{-3}$ with a total lateral variance of about 15 kg m$^{-3}$. At 110 km depth, densities show a larger variance of 55 kg m$^{-3}$ around a lower mean of 3299 kg m$^{-3}$, while at 190 km depth the scatter reduces to 5 kg m$^{-3}$ for a mean of 3353 kg m$^{-3}$.

## 2.4 3D gravity modelling

We have used the above constraints on the structure and density of the sedimentary and volcanic cover (Fig. 2, Table 1, Table 2), the crystalline crust (Fig. 3, 4), and the mantle (Fig. 5) to set up an initial 3D density model. To delineate discrete density bodies, the corresponding scattered structural information has been interpolated to regular grids of 50×50 km horizontal resolution. For example, an initial depth to the top of the Basal Crustal Layer has been constructed through

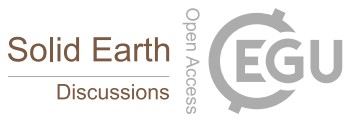

interpolation (and extrapolation) across the entire continental crustal domain of the study area using the KRISP refraction seismic data (Fig. 4). The vertical resolution of the generated 3D density model corresponds to the variable thicknesses of the different units.

In most cases, a constant density has been assigned to the modelled structural units (Table 2; Fig. 5a). For the Upper and
Basal Crustal Layers of the continental crystalline crust, the starting densities have been set to $\rho=2750$ kg m$^{-3}$ and $\rho=3000$ kg m$^{-3}$, respectively, while the oceanic crust has been assigned a value of $\rho=2900$ kg m$^{-3}$. As an exception, the mantle between 100 and 200 km depth is represented by point-wise density information (as derived from S-wave velocities; Fig. 5b-d), forming a voxel cuboid with horizontal and vertical resolutions of 50 km and 20 km, respectively.

The gravity field data that we have chosen to utilise is EIGEN-6C4, a combined surface and satellite data-based global
gravity model released by GFZ Potsdam and GRGS Toulouse (Förste et al., 2015). As we are mainly interested in the density configuration of the deeper crust, we have used the corresponding Bouguer gravity anomaly in the onshore parts of the study area, complemented by Free-air gravity anomalies in the offshore domain (Fig. 6a). Accordingly, the density model that was to be adjusted to fit the observed gravity did not include any masses above sea-level (i.e. masses within the Bouguer plate).

Based on originally constructed XYZ-grids, the gravity modelling software IGMAS+ geometrically approximates 3D density bodies by multiple polyhedra that are spanned through triangulation between 2D vertical slices (working planes). Given the N-S extension of the model area of 1100 km (Fig. 1a) and the horizontal grid resolution of 50 km, 23 E-W-striking working planes have been created for this study. IGMAS+ calculates the gravity field as the sum of the effects of all triangulated polyhedra (and voxel cuboids) while stepwise and interactive modifications of geometries and/or densities have been
performed along the 23 working planes to obtain the desired fit between modelled and measured gravity.

As the main interest of our study was to assess the density configuration of the continental crystalline crust, we have restricted modifications of the starting 3D density model to this particular structural domain. Indeed, we have found that a reasonable fit between calculated and measured gravity (Fig. 6) can be obtained when keeping the density configurations of the sedimentary and volcanic cover as well as the mantle domains fixed (Section 3). We have further improved the fit
between calculated and measured gravity by modifying the depth of the top of the Basal Crustal Layer in areas away from the gravity-independent constraints of the KRISP profiles. Secondly, the Upper and Basal Crustal Layers have been subdivided into lateral domains of different densities while taking into account the main trends in the corresponding P-wave velocity configurations (Fig. 4d, e). Finally, the Moho depth has been adjusted for a small area in eastern Kenya that lacks any gravity-independent constraints (Fig. 3a).

**3 Results: 3D density configuration of the continental crust**

By adjusting the density configuration of the sub-sedimentary continental crust within the geometrical constraints of the KRISP refraction seismic profiles, it is possible to reproduce the main observed gravity anomalies (Fig. 6a, b). Almost 90 %



of the residual gravity is within the range of ±30 mGal (Fig. 6c). Thereby, the half-wavelengths of local residual anomalies exceeding ±30 mGal are smaller than 100 km. Hence, the 3D density model best reproduces gravity anomalies of larger wavelengths.

The final density model is characterised by a continental Upper Crustal Layer that is denser in the east ($\rho$=2800 kg m$^{-3}$) than

in the west ($\rho$=2750 kg m$^{-3}$; Fig. 7a). This density difference corresponds well with higher P-wave velocities observed on KRISP line F as well as in the easternmost parts of lines D and E (Fig. 4c). Furthermore, we find that the modelled boundary between the high- and low-density domains is almost consistent with Azania's western margin (cf. Fig. 1b; Fritz et al., 2013).

Combining the gravity-constrained density and mean P-wave velocity (6325 m s$^{-1}$) of the Upper Crustal Layer in western

Kenya with the laboratory-derived property compilations of Christensen and Mooney (1995), this unit might represent granite-granodiorites, phyllites and/or paragranulites. This is fairly consistent with the lithologies of the Precambrian basement domains of western Kenya as observed at the Earth's surface (Tanzania Craton, Western and Eastern Granulites, Arabian Nubian Shield; Fig. 1b, Table 3). Both geophysical and geological data are thus indicative of a mixture of meta-sedimentary and meta-igneous rocks (Table 4). In contrast, the physical properties of the Upper Crustal Layer in eastern

Kenya (2800 kg m$^{-3}$; 6425 m s$^{-1}$) are indicative of diorites (according to the property tables of Christensen and Mooney, 1995), while the spatially corresponding microcontinent Azania is dominated by meta-igneous rocks (Table 3).

The thickness of the Upper Crustal Layer (Fig. 7a) varies between 0 km (at the COB) and >30 km in western Kenya where it is constrained by the KRISP refraction seismic lines. Based on these data, domains of reduced thickness spatially correlate with domains characterised by large depths to basement (Fig. 2a), large sedimentary thickness (Fig. 2b), and reduced total

crustal thicknesses (Fig. 3b). These domains are known to have been affected by Mesozoic and/or Cenozoic rifting.

The Basal Crustal Layer is subdivided into four domains of distinct densities (Fig. 7b). The lowest density of 2920 kg m$^{-3}$ is found underneath the Turkana and Northern Rifts (cf. Fig. 2), while the highest density of 3050 kg m$^{-3}$ has been modelled for the south-eastern parts of the study area. In the western and north-eastern parts of the study area, the modelled Basal Crustal Layer shows a density of 3000 kg m$^{-3}$. Again, these density differences correspond well with the trends in P-wave velocity

variation along the KRISP refraction seismic profiles (Fig. 4e). A comparison of the densities and mean P-wave velocity values of the Basal Crustal Layer with the property compilations of Christensen and Mooney (1995) points to an overall dominance of rocks with gabbroid composition (Table 4). Only the southeastern parts of the Basal Crustal Layer would consist of mafic granulites accordingly.

The Basal Crustal Layer is thinnest (<10 km thick; Fig. 7b) underneath the Turkana and Northern Rifts as well as the Nyanza

Trough (cf. Fig. 2a). The greatest thicknesses (>20 km) are reached east of the Southern Rift (KRISP line F) and in northeastern Kenya below the Anza Basin (cf. Fig. 2b). The modelled thickness anomalies of the Basal Crustal Layer differ significantly in wavelength (<150 km) and spatial distribution from the density configuration (four regional domains; Fig. 7b) and the Moho geometry (Fig. 3a). At the same time, large thicknesses of the Basal Crustal Layer locally correlate





with reduced thicknesses of the Upper Crustal Layer and the prevailing strike direction of these equivalent structures is WNW-ESE (Fig. 7a, b), such as beneath the axis of the Anza Basin (cf. Fig. 2a).

# 4 Implications for the strength of the lithosphere

## 4.1 Thermal and rheological modelling approach

We use the 3D density model as a basis for assessing the thermo-mechanical configuration of the lithosphere. Based on the assumption that heat is transported predominantly by conduction within the Earth's lithosphere, we numerically solve the 3D equation of heat conduction using the finite element method as implemented in the software package GMS (e.g. Cacace et al., 2010). For a thermally equilibrated system (steady-state conditions) the mathematical formulation of the relevant equation reads as:

$$\nabla * (\lambda_b \nabla T) = -S \qquad \text{(Eq. 3)}$$

where $\nabla$ is the Nabla operator [m$^{-1}$], $\lambda_b$ is the bulk thermal conductivity [W m$^{-1}$ K$^{-1}$], $T$ is the temperature [K], and $S$ is the radiogenic heat production [W m$^{-3}$].

For the calculation of the 3D conductive thermal field each model unit is assigned a constant value of radiogenic heat production and bulk thermal conductivity (Table 2, Table 4). Thereby, the thermal property values are chosen from

published compilations of laboratory measurements according to the prevailing lithologies of the model units (Table 4). To account for the depth-variable porosities of the sedimentary and volcanic rocks in the Eastern Basins domain (Appendix A), the average bulk (solid plus fluid) thermal conductivity, $\lambda_b$, is calculated for the defined depth levels (Table 2) using the geometric mean equation:

$$\lambda_b = \lambda_w^{\phi_z} * \lambda_s^{(1-\phi_z)} \qquad \text{(Eq. 4)}$$

where $\lambda_w$ is the thermal conductivity of liquid water assumed to fill the pore space [$\lambda_w$=0.6 W m$^{-1}$ K$^{-1}$), $\phi_z$ is the depth-dependent porosity and $\lambda_s$ is the thermal conductivity of the solid rock components.

The model setup is finalised by setting thermal boundary conditions. While the lateral boundaries of the model are closed to heat flow, the upper thermal boundary condition is set to a constant value of 20°C at the topography (respectively bathymetry), which represents the annual mean surface temperature of Kenya as derived from a global climatological model

(Jones et al., 1999). Finally, we define the lower thermal boundary condition by the depth of the 1350°C-isotherm (Fig. 8a) as derived from the combined S-wave velocity models (Fishwick, 2010; Adams et al., 2012) using the approach of Priestley and McKenzie (2006).

To predict spatial variations in the strength of the lithosphere of Kenya, we follow a similar approach as previous regional (e.g. Gac et al., 2016) or global studies (e.g. Tesauro et al., 2012, 2013) while using the code of Cacace et al. (2016). In

general, the strength of the lithosphere can be described as the maximum differential stress ($\Delta\sigma_{max}$) that rocks under certain P-T conditions are able to resist without experiencing brittle or ductile deformation:

$$\Delta\sigma_{max} = \sigma_1 - \sigma_3 \qquad \text{(Eq. 5)}$$

where $\sigma_1$ and $\sigma_3$ are the maximum and minimum principal stresses, respectively. Furthermore, a certain rock type will deform according to the mechanism that requires least differential stress at a given depth. At shallow depths, rocks predominantly deform by brittle behaviour, which is empirically described by Byerlee's (1978) temperature-independent law:

$$\Delta\sigma_b = f_f \rho_{bulk} g z (1 - f_p) \qquad \text{(Eq. 6)}$$

where $\Delta\sigma_b$ is the brittle yield strength [Pa], $f_f$ is the friction coefficient , $\rho_{bulk}$ is the bulk density [kg m$^{-3}$], $g$ is the acceleration due to gravity [$g$=9.81 m s$^{-2}$], $z$ is the depth below topography [m] and $f_p$ [-] is the pore fluid factor ($f_p$=0.36). Since strain is overall extensional within the EARS (e.g. Bosworth and Strecker, 1997; Stamps et al., 2014), the friction coefficient is chosen to represent extensional deformation ($f_f$=0.75).

At larger depths, if temperature is sufficiently high, rocks experience ductile deformation associated with solid-state creep (dislocation or glide). The dominant creep mechanism for the crust and upper mantle is dislocation creep, which represents temperature-dependent non-linear viscous flow (Karato and Wu, 1993). For differential stresses of $\Delta\sigma_b$>200 MPa within the mantle, Dorn's law describing solid-state creep behaviour of olivine is a better approximation of the mode of mantle rock deformation (Goetze, 1978; Goetze and Evans, 1979).

The corresponding ductile yield stress equations read as power-law rheology functions:

$$\Delta\sigma_{d,<200\,MPa} = \left(\frac{\dot\varepsilon}{A_p}\right)^{\frac{1}{n}} \exp\left(\frac{Q}{nRT}\right) \qquad \text{(Eq. 7a)}$$

$$\Delta\sigma_{d,>200\,MPa} = \sigma_D \left(1 - \left[-\frac{RT}{Q_D} ln\frac{\dot\varepsilon}{A_D}\right]^{\frac{1}{2}}\right) \qquad \text{(Eq. 7b)}$$

where $\dot\varepsilon$ is the reference strain rate [$\dot\varepsilon$=10$^{-15}$ s$^{-1}$; e.g. Sonder and England, 1986], $A_p$ is a pre-exponential scaling factor [Pa$^{-n}$], $n$ is the power-law exponent, $Q$ is the activation energy [J], $\sigma_D$ is the Dorn law stress [Pa], $Q_D$ is the Dorn law activation

energy [J] and $A_D$ is the Dorn law strain rate [s$^{-1}$], while $R$ and $T$ are the universal gas constant [R=8.314 J K$^{-1}$ mol$^{-1}$] and the absolute temperature [K], respectively.

The variation of maximum yield strength with depth for a certain XY-position is expressed by a yield-strength envelope (YSE; Goetze and Evans, 1979):

$$YSE = \min(\Delta\sigma_b, \Delta\sigma_d) \qquad \text{(Eq. 8)}$$

Based on this vertical variation in strength, we finally compute the depth-(z-) integrated strength of the entire lithosphere (respectively crust),

$$\sigma_L = \int_0^z (\sigma_1 - \sigma_3) * dz \qquad \text{(Eq. 9)}.$$

To calculate the ductile strength of the lithosphere using Eq. (7a,b), temperatures are derived from the 3D conductive thermal model. The rheological parameters assigned to the model units (type rheologies) are consistent with the physical

properties (i.e. seismic velocity, density) and derived lithologies of the units (Table 4). For example, considering the demonstrated differences between the western and eastern domains of the Upper Crustal Layer (Fig. 7a; Section 3), we have chosen "granite (dry)" and "diorite (dry)" as their rheological type compositions, respectively. In a similar way, the





rheological parameterization of the Basal Crustal Layer is also guided by prevailing lithologies as inferred from available geophysical observations (Table 4). The overall reasoning for the parameterization is that we assume that the larger the seismic velocity and density, the stiffer the crustal rheology.

## 4.2 Results

Figure 8b shows the surface heat-flux density derived from the 3D thermal model. The spatial correlation of high heat flux density in western Kenya ($>70$ mW m$^{-2}$) with shallower depths of the 1350°C isotherm (Fig. 8a) is obvious. In contrast, low heat-flux values ($<60$ mW m$^{-2}$) occur in the Eastern Basins domain where the 1350°C isotherm is located at larger depths and the sediment thickness is high (Fig. 2b).

To validate the 3D thermal model, we have analysed temperatures measured over variable depth ranges in shallow boreholes (with minimum and maximum depths below topography of 19 m and 280 m, respectively; Nyblade et al., 1990; Wheildon et al., 1994). The differences between the geothermal gradients observed in boreholes and corresponding predictions of the 3D thermal model are shown in Figure 8c. The overall range in the differences between modelled and measured gradients (with a mean of -4 K km$^{-1}$) suggests that the model neither significantly overestimates nor underestimates the heat arriving at the surface of the model. On the other hand, modelled geothermal gradients which are too low compared to the measured values are concentrated close to the Nyanza-Kenya Rift junction, while overestimated geotherms are scattered across the entire study area.

The calculated maximum differential stress varying with depth is illustrated by YSEs for four locations along an across-rift profile (Fig. 9). In western Kenya (locations A and B), the Basal Crustal Layer forms a weak domain between the Upper Crustal Layer and the mantle ("jelly-sandwich model"; Hirth and Kohlstedt 2003). Towards the east, the strength of the Basal Crustal Layer tends to increase so that the YSE at point D is "saturated" showing that ductile deformation is restricted to the mantle ("crème-brûlée model"; Jackson, 2002).

According to our calculations, the total strength integrated over the full depth of the lithosphere shows a large variability of $\sigma_L$=12.3-13.8 log$_{10}$ Pa m across the study area (Fig. 10a). Almost as large as this is the spatial variability in integrated crustal strength (Fig. 10b). Both distributions reveal the largest strengths in the southeastern parts of the study area and the smallest strengths in northern Kenya. The northern domains of crustal and lithospheric weakness correlate with the shallowest depths of the 1350°C isotherm (Fig. 8a) and the highest surface heat-flux densities (Fig. 8b). In these sectors, up to 99 % of the calculated strength is associated with the crust (Fig. 10c). In contrast, the Turkana and Northern Rift region – where the Moho is situated at shallow depths (Fig. 3a) and the crust is thinned (Fig. 3b) – bears most of its strength (up to 93 %) within the lithospheric mantle. In the Southern Rift domain (including the Tanzania Divergence), both lithospheric and crustal strengths are significantly reduced compared to the corresponding off-rift domains.



# 5 Discussion

## 5.1 Modelled density configuration of the crust

### 5.1.1 Model sensitivity and robustness

For the mantle below 100 km depth we have converted S-wave velocities into temperatures (following the empirical

approach of Priestley and McKenzie, 2006) and densities assuming that the mantle is homogeneously composed of undepleted spinel peridotite (Mechie et al., 1994). It would be highly speculative to implement lateral heterogeneities in mantle composition to assess the related influence on the gravity field. Testing alternative uniform mantle compositions for model sensitivity, however, is a straightforward undertaking as it means averaging physical properties according to alternative relative portions of constituting minerals (cf. Table A1). Such a change in the model setup results in an overall

shift of the calculated densities. For example, while for the undepleted spinel peridotite a mean of 3325 kg m$^{-3}$ is calculated, the mean density for a harzburgite composition (Irifune, 1987) is 3373 kg m$^{-3}$, and for a pyrolite composition (Irifune and Ringwood, 1987) it is 3400 kg m$^{-3}$. The spatial variances in density, however, turn out to be nearly identical ($\Delta\rho$=498±2 kg m$^{-3}$) for the three compositions, as are the standard deviations ($\rho$=22.1±0.2 kg m$^{-3}$) and the spatial distributions of density highs and lows. Most importantly, all of the temperature-controlled density variations tested result in

negligible effects on the gravity field, i.e. only ±2 mGal compared to a homogeneous mean mantle density. The modelled deeper part of the mantle (>100 km) thus does not have any significant influence on the gravity-driven investigation of the density configuration of the crust.

For mantle depths between the Moho and 100 km we have applied the linear relationship proposed by Ravat et al. (1999) to derive densities from different P-wave velocity datasets. These datasets (Fig. 5a) are remarkably consistent in terms of the

location and geometry of the LVA underneath the rift while differing in terms of velocity contrast between the rifted mantle domains and undisturbed surroundings. Our model here comprises a relatively large density contrast in accordance with the KRISP seismic velocities ($\Delta\rho\leq$25 kg m$^{-3}$; Fig. 5a). According to the tomographic studies (e.g., Achauer and Masson, 2002), the negative velocity anomaly with respect to PREM amounts to ≤12 %, which translates to $\Delta\rho$<10 kg m$^{-3}$. When decreasing the mantle-density difference between the rift and surroundings to $\Delta\rho$=10 kg m$^{-3}$, we find that the calculated gravity response

changes by up to 50 mGal, which implies a strong impact on the assessment of crustal densities. We favour, however, the presented model with a larger density contrast since it involves crustal densities that are consistent with those also derived from the KRISP seismic velocities (Eq. (1); Fig. 4d, e).

The largest uncertainties concerning the modelled depth of the Moho exist in northeastern Kenya where this information derives from a (non-unique) gravity-constrained 3D density model (Woldetinsae, 2005). The only alternative model is the

global model "Litho 1.0" (with a lower spatial resolution of 1°; Pasyanos et al., 2014). Aside from local depth differences ($\Delta z$<5 km over regions spanning <150 km) that would correspond to differences in calculated gravity of <±30 mGal, the two models agree well in terms of regional Moho trends. Hence, these local uncertainties do not question one of the main



findings of this study that northeastern Kenya is regionally underlain by a lower crust of high density ($\rho$=3000 kg m$^{-3}$) with NW-SE oriented thickness maxima (Fig. 7b).

Away from the KRISP seismic lines, it is not only the densities of the Upper and Basal Crustal Layers, but also their relative thicknesses that constitute free parameters for gravity modelling. To account for this uncertainty, we additionally present the

vertically averaged density of the crystalline crust (Fig. 7c) calculated as the thickness-weighted average density of the two crustal layers (Fig. 7a, b). Even if there is some uncertainty concerning the vertical distribution of masses within the crust (in particular in eastern Kenya), there is a very robust trend of increasing densities from western to eastern Kenya.

Finally, modelling the sedimentary and volcanic cover involves uncertainties related to assumptions on lithologies and porosities (Table 1, Table 2; Appendix A). For western Kenya, the use of information on lithologies (e.g. Morley et al.,

1992) and densities (e.g. Morley, 1999) complemented by the KRISP constraints on crustal and mantle densities directly results in a reliable fit between calculated and measured gravity (Fig. 6c). We attribute this also to the low volumes of sedimentary/volcanic rocks in western Kenya (Fig. 2b). In contrast, sediment thicknesses in the domain of the Eastern Basins are much larger, pointing also to larger porosity variations (due to differential states of compaction). Our model is consistent with the available information on porosity-controlled density increase in the Anza (Jose and Romanov, 2012) and Lamu

basins (Yuan et al., 2012). Assuming an alternative scenario of a fully compacted sediment package in the Eastern Basins domain with a homogeneous bulk density of 2710 kg m$^{-3}$, for instance, would increase the gravity response by locally up to +80 mGal. To restore the gravity fit, however, this density increase would require a reduction of the average density of the underlying Upper Crustal Layer by <40 kg m$^{-3}$. Hence, the general trend of eastward increasing densities in the entire crust (Fig. 7c) would still remain.

**5.1.2 Model interpretation**

*Upper Crustal Layer*

The 3D model reveals that the strongest density contrast within the Upper Crustal Layer largely correlates with the western margin of the microcontinent Azania as proposed by Fritz et al. (2013; Fig. 1, 7a). Azania is inferred to be separated from tectonic blocks of the Mozambique Belt (Arabian Nubian Shield and Eastern Granulites; Fig. 1b) by west-dipping thrust

faults (Fritz et al., 2013). In correspondence with this tectonic model, we interpret the slight westward offset of the modelled density contrast with respect to the surface boundary (Fig. 7a) as being due to fault dips through which Azania's margin is located farther west towards greater depths.

In contrast to previous studies (e.g. Tesha et al., 1997; Simiyu and Keller 1997) that proposed a major density difference in the upper crust between the Tanzania Craton ($\rho$=2680 kg m$^{-3}$) and the Mozambique Belt ($\rho$=2700 kg m$^{-3}$), our study does not

provide arguments for a further separation of the Upper Crustal Layer into major (density) domains (such as those of Fritz et al., 2013; Fig. 1b). Both, seismic velocity distribution (Fig. 4d) and residual gravity (Fig. 6c) only indicate lower-amplitude and, in particular, smaller-scale density variations inside each of the two (western and eastern) density domains (Fig. 7a). In contrast to Azania being predominantly composed of meta-igneous rocks, the Precambrian domains of western Kenya





comprise both meta-sedimentary and meta-igneous rocks (Table 3), which might explain the similarities of the western Kenyan domains in terms of overall velocity and density structure. Hence, we conclude that the most important physical contrast within the Upper Crustal Layer corresponds to the boundary between Azania and the Mozambique Belt.

*Basal Crustal Layer*

We have extended the Basal Crustal Layer as constrained along the KRISP refraction seismic profiles all across the study area where, however, it shows variable densities and thicknesses (Fig. 7b). For the main rift domain (with densities of $\rho$=2920 kg m$^{-3}$), there is strong consensus that the high P-wave velocities reflect mafic to ultramafic rocks that intruded and/or underplated the lower crust during Cenozoic rifting, especially if the large volumes of differentiated volcanic rocks

are taken into account (e.g. Lippard, 1973; Hay et al., 1995; Mechie et al., 1997; Thybo et al., 2000). Accordingly, the high-density crustal material is interpreted as a residue of magmatic differentiation after ponding of magma around the Moho, a process known from various continental rifts (e.g. Thybo and Artemieva, 2013). Compared to the remaining model units of the Basal Crustal Layer, the proposed mafic rocks underlying the Kenya Rift show relatively low velocities and densities (Fig. 4e, 7b), which might be due to elevated mantle and crustal temperatures (cf. Fig. 5c, 8a) and related thermal expansion

of the rocks (see also e.g. Maguire et al., 1994).

Underplating during Cenozoic rifting and variable mantle temperatures at the present-day, however, cannot explain the whole complexity of the modelled Basal Crustal Layer. The largest thicknesses of this layer, for example, have been modelled northeast of the Cenozoic rift (Fig. 7b), i.e. in a NW-SE striking band underlying the similarly oriented Anza Basin (cf. Fig. 2b). This configuration might therefore be indicative of a Cretaceous phase of extensional tectonics (Foster and

Gleadow, 1996) and magmatic underplating related to the development of this (failed) rift. On the other hand, the Basal Crustal Layer also shows considerable thicknesses beneath the Mandera and northern Lamu Basins (Fig. 7b) as well as their continuations towards Somalia where "anomalous basement" with densities of $\rho$=3015-3300 kg m$^{-3}$ is described (Rapolla et al., 1995). Hence, it is also plausible that the high-density Basal Crustal Layer underlying the Eastern Basins already had formed during the Jurassic rifting events that culminated in the formation of the Indian Ocean.

Finally, there are domains of great crustal thickness within the Basal Crustal Layer that do not spatially correlate with rifted Phanerozoic sedimentary basins. Alternative processes such as metamorphism of the hydrous crust due to pressurisation and heating (e.g. Semprich et al., 2010) could also have potentially produced such a high-density lower crust. One indication for different origins of the Basal Crustal Layer in the Southern Rift domain is provided by KRISP line G: The basal layer below the rift reveals a much stronger seismic reflectivity than domains outside the rift (Thybo et al., 2000). According to our

model, this change in reflectivity is accompanied by an increase in density and thickness towards the SE (Fig. 4c, 7b).

Within the southeastern domain of highest densities ($\rho$=3050 kg m$^{-3}$; Fig. 7b), the largest thicknesses are attained around the proposed boundary between Azania and the Mozambique Belt (Eastern Granulites; cf. Fig. 1b; Fritz et al., 2013). In northern Tanzania, this subdomain includes the Masai Plateau, a tectonic block of Neoproterozoic (Pan-African) meta-sedimentary rocks (Selby & Mudd 1965; Fig. 7a,b) that Ebinger et al. (1997) interpreted as a discrete terrane based on both distinctive





gravity and magnetic anomalies. Hence, the formation of high-density lower crust in this area might also be related to magmatic and/or metamorphic processes that accompanied the Precambrian amalgamation associated with the East African Orogen.

*Residual gravity*

Even if a better fit between modelled and measured gravity could theoretically be achieved, we refrain from implementing additional contrasts into the 3D density model because (i) the wavelengths and amplitudes of the residual gravity anomalies (<200 km, <±30 mGal; Fig. 6c) are beyond the scope of this study and (ii) the results would largely remain highly non-unique due to the scarcity of gravity-independent constraints on subsurface densities. Nevertheless, some implications on

smaller-scale density heterogeneities can be derived from the residual gravity directly.

Given the half-wavelengths in the residual gravity anomalies that predominantly remain shorter than 150 km, their causes must be located within the crust rather than deeper in the mantle. This corresponds well with local P-wave velocity anomalies as detected on some of the KRISP profiles. For instance, the residual gravity of our study reveals a positive anomaly (indicating a mass deficit in the 3D density model) on the eastern margin of the Kenya Rift northeast of the Nyanza

Rift – Kenya Rift junction (point A; Fig. 6c). KRISP line D crosses this area and reveals a 50 km wide structure at <10 km depth with velocities up to $v_p$=0.2 km s$^{-1}$ larger than the surroundings (e.g. Maguire et al., 1994; KRISP line D, Fig. 4a). Keller et al. (1994) interpreted this high-velocity structure as being caused by numerous mafic intrusions (dykes). Likewise, Prodehl et al. (1994) related a structure of high seismic velocity and reflectivity at the southeastern tip of KRISP line E (point B; Fig. 6c) to gabbroic intrusions inferred to predate Cretaceous (Anza) and Cenozoic (Kenya) rifting.

The gravity high in the Southern Rift domain (point C; Fig. 6c) has been reproduced by a modelled high-density body (with 20 km width, at 4 km depth) that is interpreted as a massive intrusion within the thick low-density tuffs and ashes of the Neogene to recent rift (Simiyu and Keller, 2001). On the other hand, the gravity high located farther southeast (point D; Fig. 6c) spatially correlates with a shallow zone of low electrical conductivity (high resistivity; <100 km wide, <10 km deep), which is bordered to the SE and NW by conductive material (interpreted to contain deposits with hot saline fluids;

Meju and Sakkas, 2007). Taken together, these observations consistently relate local gravity (residual) highs, i.e. mass deficits in the regional 3D density model, with massive and impervious high-density intrusions.

The four locations referred to (points A-D) appear as residuals of >30 mGal (Fig. 6c). Interestingly, most of such significant mass-deficit locations in our model are situated in a narrow N-S oriented band along the eastern margin of the rift and west of Azania's western margin. It is not clear whether these shallow crustal structures developed in response to Precambrian

Azania-Mozambique Belt collisional processes or during Cenozoic extension. Prodehl et al. (1994), for instance, favoured the first scenario. In any case, the related physical contrasts observed at the present-day should be taken into account when investigating local deformation such as dyke emplacement. There might be, for example, a relationship between the modelled high-density bodies and the active volcanoes of the study area, although the latter are offset from the mapped mass deficit areas (Fig. 6c).



The residual gravity also reveals areas of significant mass excess in the 3D density model, i.e. areas in which the calculated gravity overestimates the observed values by >30 mGal (Fig. 6c). Most of these spots are concentrated in the northern Kenya Rift and the Nyanza Rift. Since there is no correlation between the geometries of the rift fill (Fig. 2b) and the location of the mass excess areas, we can exclude that a modification of sediment and volcanic densities would decisively improve the

gravity fit. Sedimentary and volcanic rocks in the Nyanza Rift, for instance, are less widely distributed (Fig. 2a; Beicip, 1987) than the significant observed gravity low (Fig. 6a) and the related mass excess in the model (Fig. 6c). The size of the negative anomaly rather points to a source within the crystalline basement.

The presented model reveals mass excess (Fig. 6c) in places where the Moho is situated at relatively shallow depths (as constrained by the KRISP refraction seismic data; cf. Fig. 3a), the crust is strongly thinned (cf. Fig. 3b), and mantle

temperatures are increased (Fig. 5c; as indicated by low shear-wave velocities). Still, the spots of modelled mass excess reflect smaller-wavelength anomalies compared to these major products of rifting, which overall points to intracrustal heterogeneities. These low-density domains might be a result of local thermal anomalies (that are not integrated in the 3D model and) that induce local thermal expansion (as proposed for the Basal Crustal Layer; e.g., Maguire et al., 1994) and even partial melting (as proposed for the mantle; Mechie et al., 1994). A causal relationship between modelled mass excess and

thermal perturbations is also indicated by the active volcanoes that are predominantly situated where the crust shows local spots of lower densities (Fig. 6c). Interestingly, also the geothermal gradients, underestimated by the (purely conductive) thermal model, are indicative of advective (magma or water controlled) heat transfer around the Nyanza-Kenya Rift junction (Fig. 8c; see also Wheildon et al., 1994).

An alternative explanation for low densities in the Nyanza Rift crust, however, could be seen in compositional variations.

This part of the rift is known for its exceptional Cenozoic volcanic assemblages containing large amounts of carbonatites (Jones and Lippard, 1979; Onuonga et al., 1997). A compositional rather than thermal effect would explain better the significant drop of P-wave velocity (from $v_p \approx 6.325$  to 6.150 km s$^{-1}$) in the Upper Crustal Layer at the eastern end of the Nyanza Rift (KRISP line ABC; Fig. 4d).

### 5.2 Modelled strength configuration

### 5.2.1 Model sensitivity and robustness

The rheological configuration of the lithosphere is mainly controlled by its thermal state (Eq. (7a,b)) which is assessed based on two strong assumptions: (i) heat is transported solely by thermal conduction and (ii) the modelled system is in thermal equilibrium (steady state). Previous studies have shown that thermal diffusion is the dominant heat transport mechanism in the lithosphere and thus controls the long-wavelength temperature pattern within the crust (e.g. Pollack et al., 1993;

McKenzie et al., 2005; Scheck-Wenderoth et al., 2014). The Kenya Rift, however, is well-known for its active magmatic and hydrothermal systems leading to locally perturbed surface heat flow (e.g. Nyblade et al., 1990; Ogola et al., 1994; Wheildon et al., 1994).



Furthermore, we have derived the lower thermal boundary condition (1350°C-isotherm; Fig. 8a) from mantle S-wave velocities using an empirical conversion approach (Priestley and McKenzie, 2006) and assuming that it is in equilibrium with the prescribed surface temperatures (upper boundary condition). Such modelled steady-state conditions do not account for the likelihood that the excess heat related to the mantle thermal anomaly emplaced some 45 Ma ago has not yet been

conducted to the surface due to the low thermal diffusivity of rocks (Appendix D; e.g. Wheildon et al. 1994). Consequently, it must be expected that the thermal model overestimates temperatures in the shallower parts of the lithosphere (i.e. where the thermal anomaly has not yet diffused to).

Indeed, we find modelled thermal gradients that are too high compared to measured values spread all across the study area (Fig. 8c). In contrast, however, the largest misfits are indicative of underestimated temperatures being concentrated in the

Nyanza and Kenya Rifts that are believed to be perturbed by hydrothermal activity (e.g. Wheildon et al., 1994). The misfits generally tend to decrease with increasing distance from the rift. Hence, deeper temperature measurements that are less prone to advective thermal perturbations would be a more useful basis to validate the 3D thermal model than the shallow geothermal gradients. Given the lack of such data and our interest in the crustal and lithospheric-scale thermal and rheological state of the system, we anyhow regard the conductive 3D thermal model as a suitable approximation.

The calculated lithospheric strength configuration (Fig. 9, 10) is based on a spatially invariable strain rate (of $\dot{\varepsilon}=10^{-15}$ s$^{-1}$). We are aware that estimations of the present-day strain-rate variations show a range of $\approx10^{-15}$-$10^{-18}$ s$^{-1}$ for the EARS with significant strain localisation along its rifts (e.g. Stamps et al., 2014; Melnick et al., 2015). However, we do not intend to simulate present-day deformation with these calculations (which would require a dynamic instead of a steady-state approach). Using a spatially invariant strain rate, however, allows us to uncover rheological discontinuities inherent in the

thermal state and the compositional heterogeneity of the system. Thus we provide a model mimicking conditions that potentially have controlled the rift localisation process. The resulting pattern of strength variations thereby does not change significantly when applying alternative strain rate values. For example, a strain rate of $\approx10^{-16}$ s$^{-1}$ would result in a range of lithospheric strength of $\sigma_L$=11.9-13.4 log$_{10}$ Pa m (compared to $\sigma_L$=12.0-13.5 log$_{10}$ Pa m for $\approx10^{-15}$ s$^{-1}$) while showing the same spatial trends.

The gravity-constrained 3D structural model provides the basis and thus the spatial resolution for the thermal and rheological calculations. Each model unit is populated with homogeneous average rock properties according to its prevailing lithologies (Tables 1, 2, 3, 4). These lithologies, in turn, have been inferred from gravity-constrained densities in combination with seismic P-wave velocities (Christensen and Mooney, 1995). For the Upper Crustal Layer this approach is confirmed relatively well by the consistency between the derived lithologies and geological outcrop data. A more detailed and thus

more realistic differentiation of rheological heterogeneities in the lithosphere would require even more observations. At this stage of investigating the greater Kenya Rift by means of the currently available data, however, we present only one scenario designed to reflect the main compositional trends observed by correlating high (low) density with strong (weak) rheology.

A potential key to evaluate the uncertainties inherent in the overall rheological modelling approach is provided by local observations on seismicity. Assuming that short-term deformation reflects the long-term mechanical properties of the





lithosphere, the relative abundance of earthquakes is supposed to be related to the yield strength at depth (Ranalli, 1995, 1997). Albaric et al. (2009) presented the depth-frequency distribution of earthquakes for four localities in the Southern Rift domain including the Tanzania Divergence (E-H; Table 5; Fig. 10a). At these points, the numbers of earthquakes strongly vary with depth, delineating one or more depth levels of increased seismicity. Albaric et al. (2009) interpreted these peak seismicity depths as indicating tops of brittle-ductile transitions (BDTs) and thus rheological discontinuities. For point G located in the Southern Rift (Fig. 10a), none of the modelled brittle-ductile transitions (neither in the crust nor in the mantle) fits with the observed peak seismicity depth (Table 5). In contrast, for points E and F certain peak seismicity depths correlate remarkably well with the tops of BDTs in the crust as predicted by our model. For point H in the Tanzania Divergence it is the depth of the modelled mantle BDT that is similar to the observed peak seismicity depth in the mantle. Hence, despite the various model uncertainties we identify consistencies between the modelled rheological configuration of the lithosphere and first-order observations on seismicity.

### 5.2.2 Model interpretation

We find that the integrated strength of the lithosphere varies considerably across the study area ($\sigma_L$=12.0-13.5 $\log_{10}$ Pa m; Fig. 10a) showing almost the same range of orders of magnitude as lithospheric strength distributions calculated for global models (e.g. $\sigma_L$=12.3-14.1 $\log_{10}$ Pa m under compression; Tesauro et al., 2012). Not surprisingly, this large variability in modelled strength is related to the mantle thermal anomaly (as a shallow 1350°C-isotherm (Fig. 8a) correlates with domains of low mechanical strength). On the other hand, the crust contributes an important fraction to the total lithospheric strength (Fig. 10c) and boundaries between crustal domains with different densities (and strengths) (Fig. 7a, b) can clearly be traced in the lithospheric strength distribution (Fig. 10a). Hence, according to our model, rheological differences within the crust are largely controlled by crustal composition. Relative to the thermal impact, this inherited compositional effect on crustal strength has probably been even larger in the past when the diffusion process induced by the thermal anomaly was even less advanced.

As discussed above, the differences in crustal structure and composition between western and eastern Kenya can be ascribed to (i) the Precambrian amalgamation of eastern Kenya (Azania) associated with the East African Orogeny and (ii) rifting-related upper crustal thinning and magmatic underplating in the Mesozoic. In line with the latter and based on seismicity-derived rheological models, Albaric et al. (2009) conclude that the lower crust in the southwestern parts of the study area is generally enriched in magnesium and iron ("mafic") and may be a product of magmatic events that repeatedly affected the crust since at least ≈2.5 Ga (Halls et al., 1987; Ashwal and Burke, 1989). By means of 3D gravity modelling we detected a NNE-SSW striking line as the strongest contrast in average crustal density (Fig. 7c) which might reflect this repeatedly activated tectonic zone.

Variations in lithospheric strength west of this boundary (Fig. 10a, b) are mainly controlled by variable temperature (Fig. 8a) and differences in the thickness of the (highly radiogenic; Table 4) Upper Crustal Layer (cf. Fig. 7a), while the Basal Crustal Layer is weak in western Kenya (Fig. 9, points "A", "B"). According to the model, the Upper Crustal Layer is





compositionally homogeneous across western Kenya. This is consistent with small variabilities in bulk crustal Poisson's ratios ($v$=0.25-0.26 for different Precambrian terranes of western Kenya) as derived from receiver function data (Tugume et al., 2013). These authors further propose that the different terranes due to their similar (felsic to intermediate) compositions have not exerted any major control on the localisation of Cenozoic rifting; instead, variations in the lithospheric mantle

composition are put forward as a factor. This hypothesis contrasts with the results of previous studies that emphasize the spatial correlation of both the western and the eastern branches of the EARS with Proterozoic mobile belts surrounding the Archean Tanzania Craton. Accordingly, the localisation of the rift has been related to differences in crustal composition (e.g. McConnell, 1972), crustal composition and structure (e.g., Smith and Mosley, 1993; Hetzel and Strecker, 1994), crustal thickness (e.g., Tesha et al., 1997), or lithospheric rigidity (e.g., Nyblade and Brazier, 2002) between the Tanzania Craton

and the Mozambique Belt.

Koptev et al. (2015) conceptually implement the broad low-velocity anomaly observed in the deeper mantle beneath the Tanzania Craton (Nyblade et al., 2000; Adams et al., 2012) into thermo-mechanical forward numerical experiments that reproduce how the mantle plume beneath East Africa rises beneath the craton, is deflected by the cratonic keel, and produces a magma-rich rift on its eastern boundary. Ashwal and Burke (1989) propose that the lithospheric mantle beneath the craton

is depleted, while the mantle beneath the rift is fertile (due to Precambrian collisional and post-collisional processes), which facilitated the extraction of magmas and the localisation of Cenozoic volcanism. According to our present-day 3D model, the plume-related lithospheric thinning would have been taking place beneath a compositionally heterogeneous crust (Fig. 7; Table 4). Furthermore, crustal thinning obviously focussed within the southward tapering Arabian Nubian Shield (Fig. 1b) as the easternmost part of the rheologically weaker domain of western Kenya (Fig. 10a, b). Hence, strain localisation (induced

by tensional stresses) must have been facilitated by pre-existing contrasts in rheological properties between western and eastern Kenya.

The model considering plume-craton interactions (Koptev et al., 2015) does not account for the observation that there is a mantle thermal anomaly with a larger N-S extent, underlying all of East Africa (e.g. Hansen et al., 2012), producing hotspot tectonism in Ethiopia and Kenya (e.g. Nyblade, 2011; Bastow et al., 2011) and being responsible for the higher mantle

temperatures in northern Kenya (e.g. Fig. 8a, b). New thermochronological data indicate that the northward movement of the African lithosphere with respect to the plume during the past 35 Ma (e.g., Ebinger and Sleep, 1998; Moucha and Forte, 2011) was accompanied by diachronous, spatially disparate, and partly overlapping extension along the Kenya Rift (Michon, 2015; Torres Acosta, 2015). In general, there is a prominent correlation between the N-S striking S-wave velocity low in the mantle (Fig. 5b) and the N-S striking Kenya Rift (Fig. 2, 3b). In northern Tanzania, however, the rift changes direction to

NNE-SSW, thus deviating from the N-S oriented mantle anomaly. Furthermore, due to a thick high-density lower crust of mafic-granulite composition, the rheological model predicts high crustal and lithospheric strengths for the southeastern parts of the study area, just flanking the Tanzania Divergence in the east (Fig. 10 a, b). Hence, we conclude that a further southward propagation of the rift has been prevented by this crustal domain of increased strength leading to the observed southwestward turn (or "divergence") of the rift structures right into mechanically weaker parts of the mobile belt. This is





consistent with the findings of Ebinger et al. (1997) and Le Gall et al. (2008) who interpret the relatively unfaulted and amagmatic Masai micro-block (Fig. 7a,b, 10b) to represent a cratonic fragment (beneath thin-skinned nappes) that formed a significant barrier to rifting. The Pangani Rift, which is oriented NW-SE, forms the eastern border of the Masai Block (Fig. 10b), is older (ca. 2 Ma) than the centre of the Tanzania Divergence (ca. 1 Ma; Dawson, 1992) and is less seismically

active than other parts of the Tanzania Divergence (Ebinger et al., 1997; Foster et al., 1997), however, cannot be correlated with a corresponding zone of lithospheric or crustal weakness predicted by our 3D model.

According to the model, active volcanoes within the Kenya Rift are situated where the crustal strength is lowest (Fig. 10b) which correlates with locations of an extremely thinned Upper Crustal Layer (Fig. 7a). In the Northern Kenya Rift, this narrow zone of strongest crustal thinning and volcanism is locally offset from the rift centre towards the eastern boundary of

the surface expression of the rift. This eastward shift of the volcanic chain, in turn, seems to be related to a local high in total lithospheric strength covering the rift centre (Fig. 10a). High lithospheric strength there is caused by mantle rocks that are situated at relatively shallow depths – as indicated by the corresponding Moho high (Fig. 3a) – and thus are colder and mechanically stronger than rocks in the surroundings.

The modelled strength configuration obviously provides explanations for the spatial distribution of in-rift volcanoes in the

study area. For off-rift volcanism such relationships are not that straightforward, but it seems that these volcanoes occur where gradients in lithospheric strength are large (Fig. 10a, b). Further, obviously they flank the region that is most strongly affected by the mantle thermal anomaly (Fig. 8a) and hence most severely weakened. Previous models explaining off-rift volcanism alongside the Kenya Rift involve the mechanical loading of homogeneous crust, such as the model of Ellis and King (1991) involving dilatational strain at the base of the crust of rift flanking footwall blocks as a flexural response to

normal faulting or the model of Maccaferri et al. (2014) that relates the deflection of ascending magma-filled dykes to changes in the stress field as imposed by rifting-related crustal unloading. In contrast to these models, the presented data-driven 3D model includes rheological heterogeneities within the crust and thus reveals another potential controlling factor for localised dyke propagation.

## 6 Summary and conclusions

We determined a density configuration for the crystalline crust of the greater Kenya Rift region by integrating (i) lithology-constrained densities for the sedimentary and volcanic deposits, (ii) densities derived from P-wave velocity models for the mantle down of 100 km depth, (iii) densities derived from S-wave velocity models for the mantle at 100-200 km, and (iv) in particular the gravity field. This 3D density model is consistent with the main trends in crustal P-wave velocities revealed by the KRISP refraction seismic profiles. Furthermore, we find that

• mantle density variations below 100 km depth (derived from S-wave velocity models) do not decisively affect the distribution of gravity anomalies;



- the plume-related lateral variability in mantle density between the Moho and 100 km depth amounts to $\Delta\rho \approx 20$ kg m$^{-3}$ (which corresponds to a temperature difference of $\approx 200$°C according to Eq. (B2));

- there is an overall trend of increasing mean crustal densities from mainly <2880 kg m$^{-3}$ in western Kenya to >2880 kg m$^{-3}$ in eastern Kenya, which is likely due to compositional differences;

- measured gravity anomalies larger than 100 km (half-wavelength) can be reproduced by a model comprising a two-layered crust, with both layers being laterally differentiated into domains of different densities;

- the strongest density contrast modelled for the Upper Crustal Layer corresponds with the Precambrian boundary between the Mozambique Belt in the W (known to be made up of meta-sedimentary and meta-igneous rocks) and the microcontinent Azania in the E (containing predominantly meta-igneous rocks);

- the Basal Crustal Layer reveals largest thicknesses and densities (i) underneath the Eastern Basins domain (in particular the Anza Basin) where it might have formed through magmatic underplating during Mesozoic rifting phases and (ii) beneath the Eastern Granulites where it might derive from magmatic and/or metamorphic processes accompanying the Precambrian tectonic amalgamation associated with the East African Orogen;

- there might be additional magnesium and iron rich (mafic) intrusions in the crust that are not implemented in the model but indicated by local mass deficits (positive gravity residuals >+30 mGal at half-wavelengths of <100 km);

- local areas of significant mass excess in the final density model (indicated by gravity residuals of <-30 mGal) are concentrated in the northern Kenya Rift and the Nyanza Rift where they might be related to positive thermal anomalies within the crust involving partial melting and/or rock expansion.

Having assessed the 3D density configuration of the lithosphere, we have gone one step further and derived potential implications for its thermal and rheological state. The 3D distribution of rock types inferred from geological and geophysical observations thereby has provided the basis to parameterize model units with rock physical properties. The lower thermal boundary condition has been defined as the 1350°C-isotherm as derived from S-wave velocity models (Section 2.3).

- Although the model only accounts for conductive heat transport and despite uncertainties in the assigned thermal properties, measured near-surface geothermal gradients are largely reproduced with a misfit of $<\pm10$ K km$^{-1}$ for sites spread all over Kenya and northern Tanzania.

- The overall dominance (65%) of sites with overestimated thermal gradients, however, might be related to the modelled steady-state conditions not taking into account that the mantle thermal anomaly most likely has not yet fully propagated from the mantle up to the Earth's surface.

- Around the Kenya-Nyanza Rift junction, the purely conductive thermal model significantly underestimates observed geothermal gradients pointing to advective (water and/or magmatism controlled) heat transport.



By integrating the modelled thermal field and type rheologies consistent with lithological interpretations for the modelled units, we have calculated the lithospheric yield strength configuration (as a multi-1D approach for steady-state conditions). We have assumed that the higher the seismic velocity and density of a model unit, the stiffer it is.

- First-order observations on seismicity (i.e. depths of peak seismicity) in western Kenya and northern Tanzania are strikingly consistent with the modelled strength configuration (i.e. tops of the brittle-ductile transitions).

- The model predicts smaller depth-integrated strengths for western Kenya (including the rift) than for eastern Kenya (on both crustal and lithospheric scales).

- Since the most significant strength contrast correlates with the western margin of Azania (Upper Crustal Layer) and a strong thickness increase of the Basal Crustal Layer, we conclude that the present-day rheological configuration traces back to (i) the Precambrian amalgamation associated with the East African Orogen and (ii) magmatic processes (probably underplating) affecting eastern Kenya during Mesozoic rifting phases.

- In northern Kenya the mantle thermal anomaly strongly weakens the lithosphere, while this thermal effect decreases towards the south due to a narrowing of the plume beneath the rift proper.

- According to the 3D model, plume-related lithospheric thinning has been taking place beneath a compositionally heterogeneous crust and crustal thinning concentrated within the southward tapering Arabian Nubian Shield located adjacent to the rheologically stronger domains of eastern Kenya.

- The influence of crustal heterogeneities on rift localisation during Paleogene times has probably been even stronger (relative to thermal effects) since the utilised steady-state thermal model seems to overestimate the thermal anomaly in the crust.

- Despite an overall N-S oriented mantle thermal anomaly, the western rifts of the Tanzania Divergence strike NNE-SSW, which can be explained by a domain of increased crustal strength in SE Kenya due to which the localisation of extension was deflected into a weaker domain farther west.

- The spatial correlations of in-rift volcanoes with lowest crustal strengths and off-rift volcanoes with large gradients in lithospheric strength provide new starting points for investigating volcano-tectonics and dyke emplacement in the region.

- The steady-state rheological model provides a detailed framework for future studies on dynamic processes such as rift localisation and propagation in the region.



## 7 Appendices

### Appendix A: Lithology-driven modelling of the density configuration of the sedimentary and volcanic rocks

### Eastern Kenya

Deposits of the Lamu Basin are lithologically described as a repetitive sequence of mainly siliciclastic rocks (sandstones,

siltstones, shales) and intercalated limestones (Nyagah, 1995; Table 1). Direct evidence on the density configuration of the basin infill is provided by the study of Yuan et al. (2012) who jointly investigated reflection seismic and gravity data from the central Lamu Basin. According to these observations, density increases with depth and age of the depositional sequences (the main ones of which are of Cenozoic, Cretaceous, Jurassic and Permian/Triassic ages).

Except for some Permo-Triassic evaporitic series in the Mandera Basin (e.g. Ali Kassim et al., 2002) and Miocene to

Quaternary volcanics in the northwestern Anza Basin (e.g. Class et al., 1994), also these two basins primarily contain siliciclastic rocks and limestones (Table 1). Due to this similarity in lithological trends and a lack of more detailed information on the spatial configuration of lithologies and densities, we have modelled the Mandera, Lamu and Anza Basins as one consistent domain, referred to as the Eastern Basins domain (Fig. 2b).

The downward density increase observed in the Lamu Basin obviously results from compaction and related porosity loss

with increasing burial depth and time (Yuan et al., 2012). Similar depth-dependent porosity and density trends can be assumed to characterise the sequences of the Mandera and Anza Basins that also show considerable maximum burial depths of >9 km (Fig. 2a). One commonly used relationship (Athy, 1930) to empirically describe porosity $\phi_z$ as a function of hydrostatic depth $z$ reads as

$$\phi_z = \phi_0 * e^{-cz} \quad \text{(Eq. (A1))},$$

where values for the surface (respectively depositional) porosity $\phi_0$ and the compaction coefficient $c$ vary with lithology (e.g. Hantschel and Kauerauf, 2009; Allen and Allen, 2013). We use the resulting depth-dependent porosity $\phi_z$ together with the density of the pore-filling fluid ($\rho_{por}$=1030 kg m$^{-3}$) and the lithology-dependent matrix density $\rho_{mat}$, to calculate the bulk rock density $\rho_{bulk}$ configuration in the Eastern Basins domain:

$$\rho_{bulk} = (1 - \phi)\rho_{mat} + \phi\rho_{por} \quad \text{(Eq. (A2))}.$$

For setting up the starting density model we have chosen parameter values (i.e. $c$=0.4 km$^{-1}$; $\phi_0$=0.4; $\rho_{mat}$=2720 kg m$^{-3}$) that are representative for a mixture of siliciclastic rocks and limestones and, as shown by Meeßen (2015), reasonably well reproduce the observed density increase in the Lamu Basin (Yuan et al., 2012). For the lithospheric-scale 3D gravity modelling, we have used a simplified representation of the modelled continuous density increase with depth: we have subdivided the infill of the Eastern Basins into a vertical succession of six layers (Table 2) each of which is representative of

a depth interval of up to 2 km thickness (levels A-F) and characterised by a constant average density (between 2270 and 2710 kg m$^{-3}$).

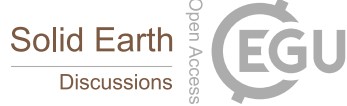

### Western Kenya

In western Kenya, as a result of spatially and temporally varying depositional environments and types of volcanism, a large variety of rock types is observed in the different rift segments. Hence, the starting 3D density model differentiates six domains (Table 1; Fig. 2b): the Lotikipi Plain domain in NW Kenya, three domains along the main Cenozoic rift and two
domains encompassing major volcanic edifices of the region.

The lithology of the deepest successions in the Lotikipi Plain domain is poorly constrained, but Tiercelin et al. (2012) have suggested that sandstones equivalent to the Late Cretaceous Turkana Grits plus lacustrine deposits form a layer of ≤700 m. These sedimentary rocks are overlain by a volcanic layer of Oligocene basalts (with interbedded tuffaceous sediments) and Miocene to Lower Pliocene rhyolitic flows (Morley, 1999). The shallowest strata are formed by upper Pliocene to recent
volcanic-derived fluvial and lacustrine sediments with a thickness of about 1000 m (Tiercelin et al., 2012; Morley, 1999). According to the generalised stratigraphy and gravity-constrained density proposed by Morley (1999), we have differentiated two layers for the Lotikipi Plain, each with a homogeneous density in the starting model (Table 2). The upper layer represents 1000 m of predominantly volcanic-derived sediments with an average density of 2350 kg m$^{-3}$. The remaining space down to the basement (with <2600 m thickness) is modelled with a density of 2550 kg m$^{-3}$ representing a mixed layer
of volcanics and clastic sediments.

The Turkana Rift domain encompasses the Turkana, Lokichar, and North Kerio Basins (Fig. 2a). The deeper infill of these basins is dominated by a variety of siliciclastic rocks (conglomerates, shales and partly arkosic sandstones; e.g. Feibel, 2011). Hence, we have used the same approach as applied to the Eastern Basins domain, i.e. approximating depth-dependent porosity and density with a vertical succession of model layers (levels B-D; Table 2). For the model layer representing the
uppermost 2 km (level A), however, a slightly higher average density of 2400 kg m$^{-3}$ has been chosen because of the abundance of massive Miocene volcanics at these depths (e.g. Morley et al. 1992).

The Northern Rift domain is segmented into the Suguta Trough, the South Kerio Trough, and the Baringo Basin (Fig. 2a,b). The sedimentary and volcanic basin infill shows maximum thicknesses of >4000 m in the Baringo Basin. Siliciclastic sediments and mafic to intermediate volcanics are the dominating lithologies (Table 1). Combined seismic and gravity
studies provide indications on the density configuration of the Baringo Basin infill, which is given as a range from 2460-2600 kg m$^{-3}$ by Swain et al. (1981) and 2460-2750 kg m$^{-3}$ by Maguire et al. (1994). For the shallower South Kerio Basin, Mugisha et al. (1997) estimated densities of 2000-2450 kg m$^{-3}$. For the starting density model, the Northern Rift domain is represented by one continuous model unit with an average density of 2550 kg m$^{-3}$.

We have combined the Nyanza, Central and Magadi Troughs together with the Tanzania Divergence into the Southern Rift
domain (Fig. 2a,b). In this domain, the thickness of sedimentary and volcanic rocks does not exceed 4200 m. The basins are mainly filled by mafic to intermediate volcanics deposited as tuffs and intercalated with volcanic-derived sediments (Table 1). Due to the high porosity of these tuffs and sediments we followed the approach of Simiyu and Keller (2001) and modelled this domain as a continuous unit with a density of 2400 kg m$^{-3}$.





The stratovolcanoes Mt Elgon, Mt Kenya and Kilimanjaro overlie the pre-Mesozoic basement (e.g. Beicip, 1987) and are differentiated in the model as the Volcanics domain (Fig. 2b). These volcanic edifices are composed of variable lithologies (Table 1). However, since they are entirely positioned above sea level and we model Bouguer anomalies induced by densities below sea level (section 2.4), their variable densities do not affect the gravity calculations.

## Appendix B: Comment on the usage of a modification of Birch's (1961) law

We have modified the widely used Birch's (1961) empirical law for crustal velocity-density relations to Eq. (1) by changing the added term from a value of 252 kg m$^{-3}$ to 350 kg m$^{-3}$. The reason for this modification can be illustrated by comparing the different densities resulting from the two equations. For a velocity of $v_p$=6.325 km s$^{-1}$, i.e. an average for the Upper Crustal Layer in the W, the corresponding densities would be $\rho_c$=2650 kg m$^{-3}$ for Birch's (1961) law and $\rho_c$=2750 kg m$^{-3}$ for its modification (Eq. (1)). We prefer the latter result since it is closer to the densities expected from the exposed (widely metamorphic) basement rock types (Table 3).

## Appendix C: Conversion of S-wave velocity derived temperatures to density

To account for the gravity effects of the mantle, we have analysed S-wave velocity data (Adams et al., 2012; Fiswick, 2010) and, in a first step, converted them into temperatures by using the empirical approach of Priestley and McKenzie (2006). In a second step, we have assessed the mantle density configuration by using the relationship between the density of a mineral ($\rho_{i,0}$ [kg m$^{-3}$]) at standard temperature and pressure conditions ($T_0$ [K]; $P_0$ [GPa]), its thermal expansion coefficient ($\alpha$ [K$^{-1}$]) and bulk modulus ($K$ [GPa]) as well as its density at in-situ temperature ($T$ [K]) and pressure ($P$ [GPa]):

$$\rho_i(P,T) = \rho_{i,0}\left[1 - \alpha_i(T - T_0) + \frac{P-P_0}{K_i}\right] \qquad \text{(Eq. (C1); e.g. Goes et al., 2000)}$$

The mineral properties ($\rho_{i,0}$, $\alpha_i$ and $K_i$) are derived from compiled laboratory measurements (Table C1). To account for the composite mineralogy of the mantle, we have averaged the in-situ densities of different minerals to obtain the bulk density $\rho_{bulk}$:

$$\rho_{bulk} = \sum x_i \rho_i \qquad \text{(Eq. (C2)),}$$

where $x_i$ is the volumetric proportion of the mineral $i$.

Mechie et al. (1994) proposed a mantle composition for the Kenya Rift and its eastern flank based on the combined analysis of P- and S-wave seismic velocities and compositions of mantle xenoliths brought up by Quaternary volcanics (Henjes-Kunst and Altherr, 1992). Accordingly, we assume that the study area is underlain by a compositionally homogeneous mantle representing undepleted spinel peridotite (Table C1).

The in-situ pressure (vertical load $P_{lith}$; Eq. (C3)) has been assessed by integrating the density-($\rho$)-controlled linear relationship between pressure and depth (below topography, z), while considering the acceleration due to gravity ($g$=9.81 m s$^{-2}$):

$$P_{lith} = g \int_0^z \rho(z)\, dz \qquad \text{(Eq. (C3)).}$$





Without knowing crust and mantle densities a priori, we have approximated in-situ pressure conditions for the mantle by taking into account the Moho depth (Fig. 3a) as well as average densities for the crust (2810 kg m$^{-3}$ as derived from KRISP) and the mantle (3300 kg m$^{-3}$; according to Ravat et al., 1999).

The involved inaccuracy of the pressure calculation (Eq. (C3)) related to the utilisation of constant average densities for the
mantle and the crust does not significantly affect the density calculation (Eq. (C1)), which can be shown by a simple scenario: Not considering a lateral variability in mantle density of 100 kg m$^{-3}$ (representing, for example, a lateral change from 3250 to 3350 kg m$^{-3}$) for the depth interval of 100-200 km would mean imposing an error of ≈0.1 GPa in the calculated pressure (vertical load; $P_{lith}$; Eq. (C3)). Since typically $K$>100 GPa (Table C1), the uncertainty in the pressure term of Eq. (C3) related to an error of ≈0.1 GPa would thus be <0.1 %. For this reason, we regard spatial density variations occurring
in the mantle and the crust as negligible for the pressure and final density calculations.

**Appendix D: Comment on the characteristic time scale of thermal diffusion**

The 1D instantaneous cooling of a semi-infinite half space (with no internal heating) is approximated by an error function

$$T_t = T_0 \cdot \mathrm{erf}(\frac{z}{2\sqrt{\kappa t}}) \qquad \text{(Eq. (D1))}$$

with $T_t$ being the temperature at time $t$ and depth $z$, $\kappa$ being the thermal diffusivity of the rocks [m$^2$ s$^{-1}$] and $T_0$ being the
initial temperature (e.g. Turcotte and Schubert, 2014). Accordingly, the amount of time t necessary for a change in $T$ to propagate a distance $l$ reads as:

$$t = \frac{l^2}{\kappa} \quad \text{(Eq. (D2))}$$

The minimum depth of the 1350°C-isotherm in the study area (at the present day) is ≈63 km (Fig. 8a). Given a thermal diffusivity of $\kappa$=10$^{-6}$ m$^2$ s$^{-1}$ (which is a typical average value for the lithosphere), a change of temperature at a depth of 63 km
would take ≈129 Ma to reach the Earth's surface. For this reason, it is very likely that the mantle thermal anomaly emplaced below the Kenya Rift ≈45 Ma ago is not yet in thermal equilibrium with near-surface temperatures.

**8 Competing interests**

The authors declare that they have no conflict of interest.

**9 Acknowledgements**

We want to thank Dr. Girma Woldetinsae for providing us with a numerical 3D density model of Ethiopia and northern Kenya that was developed in the frame of a study funded by the Katholischer Akademischer Ausländer-Dienst (KAAD).





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



**Figure 1: (a) Topography and bathymetry of the East African Rift System (EARS; from ETOPO1; Amante and Eakins, 2009); major faults of rift branches are from Chorowicz (2005); black frame marks the modelled study area. (b) Precambrian basement domains modified from Fritz et al. (2013); over wide parts of the region, these rocks are covered by Mesozoic-Cenozoic sediments and volcanics; solid white lines represent the KRISP refraction seismic profiles (Khan et al., 1999); distribution of oceanic crust derived from Müller et al. (2008); surface expressions of the Kenya and Nyanza Rifts (dashed lines) have been derived from Beicip (1987) and Milesi et al. (2010).**



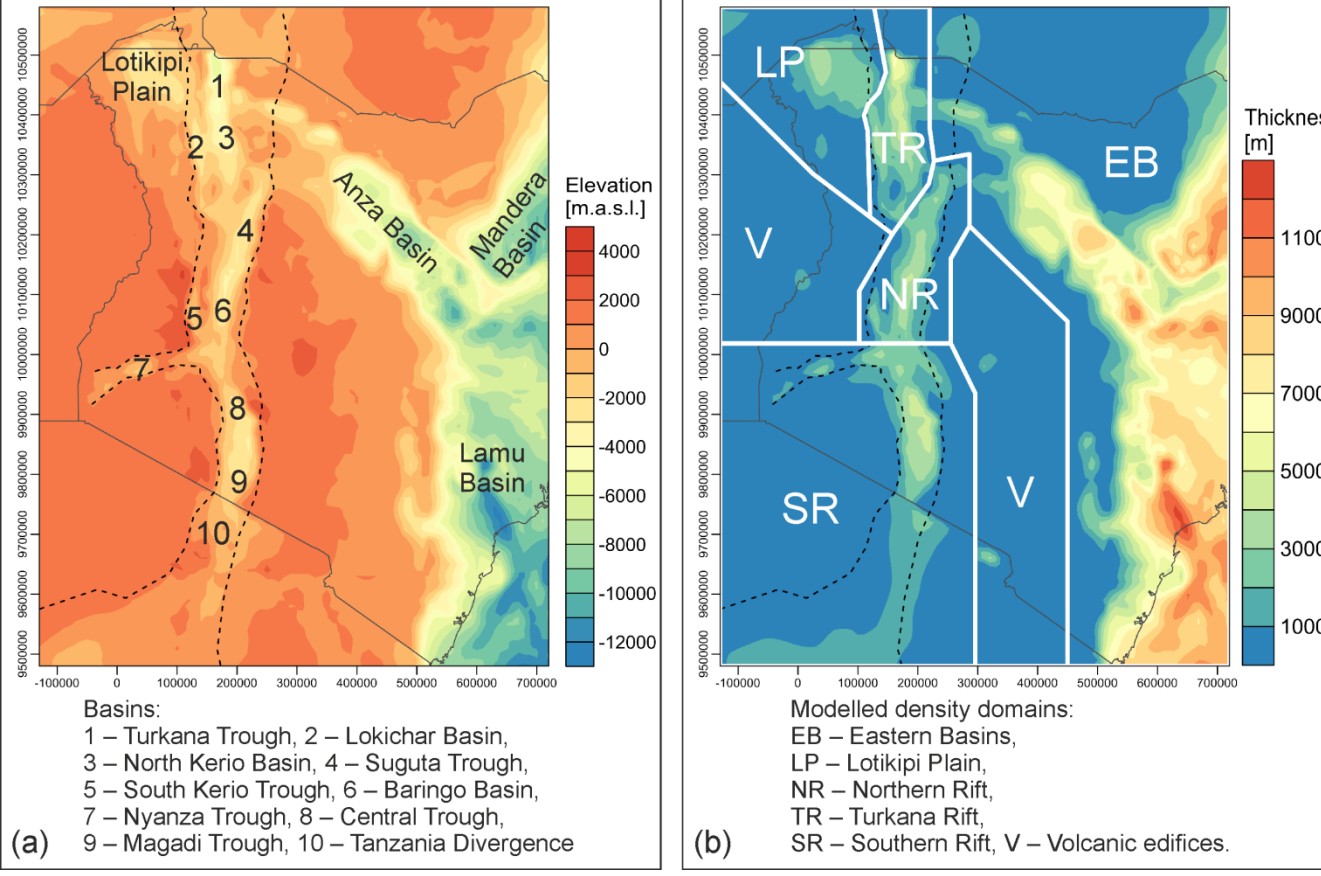

**Figure 2: Structure of the sedimentary and volcanic basin infill; all interpolations in this study have been performed using the Convergent Interpolation algorithm implemented in Petrel (©Schlumberger) and the coordinates are in UTM Zone 37S; (a) Elevation of the base of the sedimentary and volcanic rocks (i.e. top of the Precambrian basement) constructed by combining data from Kenya (Beicip, 1987) and from a newly developed global sediment thickness map; (b) Cumulative thickness of sedimentary and volcanic rocks as derived from (a) and from the topography, respectively bathymetry (ETOPO1, Amante and Eakins, 2009; Fig. 1)**





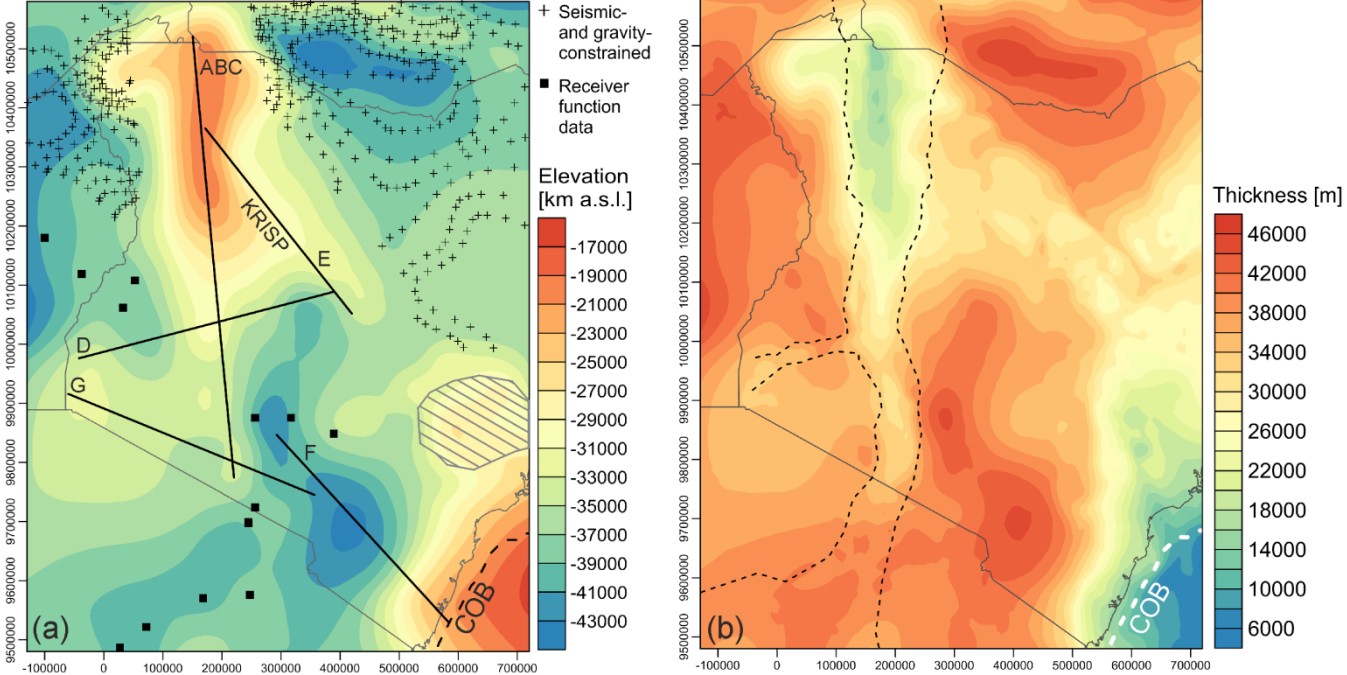

**Figure 3: (a) Crust-mantle boundary (Moho); constraining data comprise the KRISP refraction seismic profiles (A-G; Khan et al., 1999), receiver function data (squares; Tugume et al., 2012, 2013), and a 3D gravity-constrained model (crosses; Woldetinsae, 2005); the interpolation integrated also Moho depths in the oceanic crustal domain outside the study area as derived from Litho 1.0 (Pasyanos et al., 2014); the hatched area indicates where the Moho depth was interactively modified (moved upwards) according to the gravity response of the 3D density model. (b) Thickness of the crystalline crust underlying the sedimentary and volcanic deposits as obtained by subtracting the depth to basement (Fig. 2a) from the Moho in (a). COB – continent-ocean boundary.**





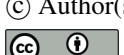



**Figure 4: Configuration of the crust and mantle derived from the KRISP refraction seismic survey (Khan et al., 1999). (a) P-wave velocity structure across KRISP profile D; the plotted velocities represent mean values for the depth interval below the respective vertex; for details on the velocity-density conversions see main text; (b) Cumulative thickness of the Upper Crustal Layer. (c) Thickness of the Basal Crustal Layer. (d) Lateral variations in P-wave velocity of the Upper Crustal Layer. Each velocity**

5  **represents a thickness-weighted vertical average of all sub-sedimentary crustal layers showing $v_p \leq 6.65$ km s$^{-1}$. Stippled line separates western Kenya with prevailing velocities of $v_{p,c} < 6.35$ km s$^{-1}$ ($\rho_c < 2760$ kg m$^{-3}$) from eastern Kenya showing mainly velocities of $v_{p,c} \geq 6.40$ km s$^{-1}$ ($\rho_c \geq 2770$ kg m$^{-3}$). Densities have been converted using Eq. (1). (e) Velocity distribution of the Basal Crustal Layer. Stippled line separates converted densities of mainly $\rho_c < 3000$ kg m$^{-3}$ in the rifted region from prevailing densities of $\rho_c \geq 3000$ kg m$^{-3}$ in the surroundings. Densities have been converted using Eq. (1).**



**Figure 5: Analysis of mantle seismic velocities; (a) Six density domains (separated by black lines; numbers in kg m$^{-3}$) have been derived from P-wave velocities for depths between the Moho and 100 km; the low-density domains underneath the rift are slightly narrower at the Moho (solid lines) than at 100 km depth (dashed lines). KRISP refraction seismic profiles A-G shown with average mantle velocity and density (converted according to Eq. 2); I – tomographic P-wave velocity model of Achauer and Masson (2002); II – tomographic model of Slack et al. (1994); III – gravity-constrained density profiles of Simiyu and Keller (1997); (b) Combined S-wave velocity datasets of "A" – Adams et al. (2012) and "F" – Fishwick (2010) (separated by the dashed line) at a depth of 150 km; (c) Temperature at 150 km depth as derived from S-wave velocity data according to the approach of Priestley and McKenzie (2006); (d) Density at 150 km depth derived from temperature according to Eq. (C1) (for details see main text).**



**Figure 6: Gravity anomalies. (a) Observed Bouguer (onshore) and Free-air (offshore) anomalies (Eigen-6C4; Förste et al., 2015).**
**This data set and corresponding reductions are available from http://icgem.gfz-potsdam.de/ICGEM/Service.html; (b) Calculated**
**anomalies of the final model; (c) Residual gravity (observed minus calculated anomalies) of the final model; black triangles mark**
**volcanoes (from the Global Volcanism Program, Department of Mineral Sciences, Smithsonian Institution, http://volcano.si.edu/).**



**Figure 7: Gravity-constrained 3D density configuration of the sub-sedimentary continental crust. (a) Thickness of the continental Upper Crustal Layer; solid black line separates domains of different average density [kg m⁻³]; dashed red line denotes the western margin of the microcontinent Azania after Fritz et al. (2013); grey line denotes spatial extension of the Masai Block as derived from Le Gall et al. (2008); (b) Thickness of the continental Basal Crustal Layer; black lines separate domains of different average density [kg m⁻³]; (c) Average density of the crystalline continental crust calculated by vertically averaging the densities of the Upper and Basal Crustal Layers according to their share in the total thickness of the crust; dashed lines delineate basement domains of Fritz et al. (2013).**





Figure 8: Modelled thermal field. (a) Depth of the 1350°C isotherm derived from S-wave velocity models (Fig. 5b) by following the approach of Priestley and McKenzie (2006) and utilised as the lower thermal boundary condition; black line and points A-D delineate the profile shown in Fig. 9; (b) Calculated surface heat flux density derived from the 3D thermal model; (c) Misfits in shallow geothermal gradients (measured minus modelled values). The measured geotherms are derived from shallow temperature data covering maximum depth ranges of 280 m (databases of Nyblade et al., 1990, and Wheildon et al., 1994).





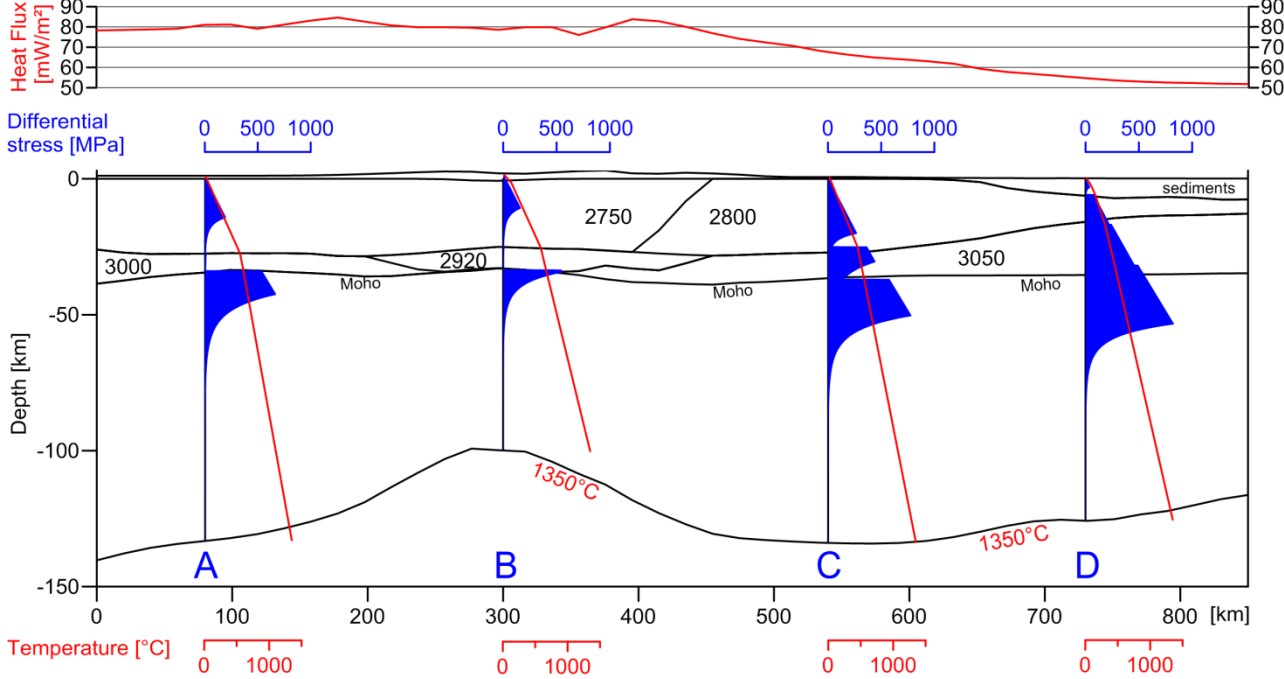

**Figure 9: Across-rift profile through the 3D structural model with yield strength envelopes at four locations (A-D; locations in Figures 8a and 10a); numbers in selected structural units indicate gravity-constrained density (kg m⁻³); also shown are the geothermal gradients for the four locations and the along-section heat flux density.**



**Figure 10: (a) Total depth-integrated strength of the lithosphere; locations of volcanoes from the Global Volcanism Program, Department of Mineral Sciences, Smithsonian Institution, http://volcano.si.edu/; white line and points A-D delineate the profile shown in Fig. 9, while points E-H mark the locations for which Albaric et al. (2009) have derived depths of peak seismicity (Table 5); (b) Integrated strength of the crust; spatial extensions of the *Masai Block* and the *Pangani Rift* derived from Le Gall et al. (2008); (c) Ratio of crustal strength with respect to lithospheric strength.**



**Table 1: Modelled lithological domains of the sedimentary and volcanic basins infill**

| Domain | Subdomains | Prevailing lithologies | Key references |
|---|---|---|---|
| V | Mt Elgon | Melanephelinites, carbonatites | Baker et al. (1987) |
| | Mt Kenya | Phonolites, trachytes, basalts | Price et al. (1985) |
| | Kilimanjaro | Alkali olivine basalts, trachybasalts/trachyandesites, trachytes, rhomb porphyries and phonolites | Williams (1969) |
| SR | **Southern Rift** (Nyanza Trough, Central Trough, Magadi Trough,  Tanzania Divergence) | Nephelinites, carbonatites, trachytes, basalts (all mainly formed as tuffs), volcanic-derived sediments, shales | **Nyanza:** Baker et al. (1971), Jones and Lippard (1979), Pickford (1982); **Central and Magadi:** Baker and Mitchell (1976), Crossley (1979), Simiyu and Keller (2001); **North Tanzania Divergence**: Ring et al. (2005) |
| NR | **Northern Rift** (Suguta Trough, South Kerio Trough, Baringo Basin) | Siliciclastic sediments (mainly sandstones), basalts, phonolites, trachytes | **Suguta Trough:** Bosworth and Maurin (1993); **South Kerio Basin:** Mugisha et al. (1997; **Baringo Basin:** Swain et al. (1981), Maguire et al. (1994), Tiercelin et al. (2012) |
| TR | **Turkana Rift** (Turkana, Lokichar and  North Kerio Basins) | Depth level A: Volcanics, volcanic-derived sediments, sandstones<br><br>Depth level B-D: Sandstones (partly arkosic), shales, conglomerates | **Turkana:** Morley etal. (1992), Ebinger and Ibrahim, 1994, Feibel (2011); **Lokichar:** Morley et al. (1992), Talbot et al. (2004), Tiercelin et al. (2004, 2012); **North Kerio:** Morley et al. (1992), Tiercelin et al. (2012) |
| LP | **Lotikipi Plain** | Depth level A: Volcanic-derived sediments<br><br>Depth level B: Basalts, rhyolites, tuffaceous sediments, sandstones (grits) | Morley (1999), Wescott et al. (1995), Feibel (2011), Tiercelin et al. (2012) |
| EB | **Eastern Basins** (Mandera, Lamu, and Anza Basins) | Sandstones, shales, limestones, silts, evaporites, volcanics | **Mandera:** Ali Kassim et al. (2002), Kerr et al. (2010); **Lamu:** Nyagah (1995), Yuan et al. (2012); **Anza:** Winn et al. (1993), Class et al (1994), Bosworth and Morley (1994) |



**Table 2: Properties of the modelled domains of the sedimentary and volcanic basins infill**

| Domain | Depth level | Maximum thickness | Modelled density | Thermal conductivity | Radiogenic heat production° |
|--------|-------------|-------------------|------------------|----------------------|-----------------------------|
| | | [m] | [kg m$^{-3}$] | [W m$^{-1}$ K$^{-1}$] | [μW m$^{-3}$] |
| V | A | 3391 | (not relevant)[#] | 2.00[§] | 1.00 |
| SR | A | 4614 | 2400 | 2.00[§] | 1.20 |
| NR | A | 4199 | 2550 | 2.30[§] | 0.90 |
| TR | A | 2000 | 2400 | 2.00[§] | 0.90 |
| | B | 2000 | 2520 | 2.20[§] | |
| | C | 2000 | 2630 | 2.40[§] | 1.20 |
| | D | 502 | 2660 | 2.50[§] | |
| LP | A | 1000 | 2350 | 1.90[§] | 1.00 |
| | B | 2562 | 2550 | 2.30[§] | 0.90 |
| EB | A | 2000 | 2270 | | |
| | B | 2000 | 2520 | | |
| | C | 2000 | 2630 | 3.00[$] | 1.10 |
| | D | 2000 | 2680 | | |
| | E | 2000 | 2700 | | |
| | F | 2225 | 2710 | | |

[#]not considered for calculations of the Bouguer gravity since the volcanic edifices are located above sea level (see main text)

[§]bulk (combined matrix and fluid) thermal conductivity, derived from Cermak and Rybach (1982)

[$]matrix thermal conductivity, derived from Midtoemme and Roaldset (1999),

[$]bulk thermal conductivity calculated using the geometric mean equation to consider porosity (e.g. Sekiguchi, 1984; Fuchs et al., 2013)

°derived from Vilà et al. (2010)





**Table 3: Precambrian basement domains of the study area**

|  | Description (Fritz et al., 2013) | Prevailing rock types | Key references |
|---|---|---|---|
| Nyanzian System of the Tanzania Craton | Craton; low-grade metamorphic Archean greenstone belt assembly | metamorphic volcanics (rhyolites, andesites, basalts), metamorphic sediments (graywackes, mudstones), meta-granites | Clifford (1970) |
| Western Granulites | Reworked pre-Neoproterozoic crust; low-grade metamorphic assemblage | paragneisses, meta-volcanosediments, quartzites, amphibolites | Mosley (1993), Maboko (1995), Möller et al. (1998), Fritz et al. (2005, 2013), Cutten et al. (2006) |
| Eastern Granulites | Part of the Eastern Granulite Cabo Delgado Nappe Complex; meta-igneous assemblage of Neo-Proterozoic juvenile crust | meta-igneous rocks (including anorthosites), meta-sedimentary rocks (including marbles) | Möller et al. (1998), Maboko and Nakamura (2002), Fritz et al. (2005, 2009), Tenczer et al. (2006) |
| Arabian Nubian Shield | Juvenile oceanic crust including numerous ophiolites, magmatic terrains | meta-volcanosedimentary sequences (mainly gneisses), ophiolites (basalts, gabbros) | Fritz et al. (2013) and references therein |
| Azania | Reworked pre-Neoproterozoic crust; microcontinent exposed e.g. in Madagascar, Somalia | orthogneisses (granite gneisses), granites, meta-sedimentary rocks | Collins and Pisarevsky (2005), Randriamamonjy (2006) |



**Table 4: Physical properties and lithologies of the model units**

| Model unit | Bulk density, $\rho$ | KRISP Mean velocity, $v_p$ | Prevailing lithology | Thermal conducti-vity, $\lambda$ | Radiogenic heat production, $A^{\S}$ | Type rheology [Reference] | Power law activation energy, $Q_P$ | Power law strain rate $A_P$, | Power law exponent, $n$ |
|---|---|---|---|---|---|---|---|---|---|
| | [kg m$^{-3}$] | [m s$^{-1}$] | | [W m$^{-1}$ K$^{-1}$] | [$\mu$W m$^{-3}$] | | [kJ mol$^{-1}$] | [Pa$^{-n}$ s$^{-1}$] | |
| Sediments, volcanics | (Table 2) | | (Table 2) | (Table 2) | (Table 2) | Quartzite, dry [1,2] | 190 | 5.00E-12 | 3.00 |
| Upper Crustal Layer, W | 2750 | 6325 | Meta-sedimentary and meta-igneous rocks | 3.0* | 1.70 | Granite, dry [1] | 186 | 3.16E-26 | 3.30 |
| Upper Crustal Layer, E | 2800 | 6425 | Meta-igneous rocks | 2.9° | 2.10 | Diorite, dry [2] | 219 | 5.20E-18 | 2.40 |
| Basal Crustal Layer, N-Rift | 2920 | 6800 | Gabbroid rocks | 2.0° | 0.35 | Diabase, dry [1] | 276 | 6.31E-20 | 3.05 |
| Basal Crustal Layer, W & NE | 3000 | 7000 | Gabbroid rocks | 2.0° | 0.35 | Diabase, dry [1] | 276 | 6.31E-20 | 3.05 |
| Basal Crustal Layer, SE | 3050 | 7000 | Mafic granulites | 2.0* | 0.15 | Mafic granulite [3] | 445 | 8.83E-22 | 4.20 |
| Oceanic crust | 2900 | NA | Gabbroid rocks | 2.6° | 0.35 | Diabase, dry [1] | 276 | 6.31E-20 | 3.05 |
| Mantle | (variable) | (variable) | Peridotite | 3.0^ | 0.01 | Olivine, dry [4] | 510 | 7.00E-14 | 3.00 |

Mantle, Dorn's dislocation glide at $\Delta\sigma \geq 200$ MPa for Olivine (dry): $\sigma_0$=8.5E9 Pa, $A_D$=5.7E11 s$^{-1}$, $Q_D$=535 kJ mol$^{-1}$

Thermal properties from: *Seipold (1992), °Cermak and Rybach (1982), ^McKenzie et al. (2005) and references therein, $^{\S}$Vilà et al. (2010)

Rheological properties from: [1] Carter and Tsenn (1987), [2] Burov et al. (1998), [3] Wilks and Carter (1990), [4] Goetze and Evans (1979)



**Table 5: Modelled brittle-ductile transition and depth of peak seismicity**

| | Modelled depth of the top of the brittle-ductile transition | | Depths of peak seismicity* |
|---|---|---|---|
| | Crust | Mantle | |
| Locations | [km] | [km] | [km] |
| E, "Bogoria" | 11.4 | 36.1 | 10.0 |
| F, "Magadi North" | 12.2 | 39.1 | 3.8; 12.2 |
| G, "Magadi South" | 12.9 | 39.4 | 20.0 |
| H, "Balangida" | 13.1 | 43.9 | 20; 40 |

*from Albaric et al. (2009)



**Table C1: Volumetric proportions of minerals and physical properties assumed to form mantle rock in the study area**

| | Volumetric proportion (Mechie et al., 1994) | Density at standard conditions for P and T | Thermal expansion coefficient | Bulk modulus |
|---|---|---|---|---|
| | | $\rho_0$ | $\alpha$ | $K$ |
| | [%] | [kg m$^{-3}$] | [$10^{-5}$ K$^{-1}$] | [GPa] |
| Olivine | 50 | $3222 + 1.182X_{Fe}$ | 2.010 | 129 |
| Orthopyroxenes | 30 | $3215 + 0.799X_{Fe}$ | 3.871 | $109 + 20X_{Fe}$ |
| Clinopyroxenes | 18 | $3277 + 0.38X_{Fe}$ | 3.206 | $105 + 12X_{Fe}$ |
| Spinel | 2 | $3578 + 0.702X_{Fe}$ | 6.969 | 198 |

$X_{Fe}=0.1$

Mineral properties as compiled in Goes et al. (2000) and Cammarano et al. (2003)