# Peer review of "The Kenya Rift revisited: insights into lithospheric strength through data-driven 3D gravity and thermal modelling"

_Solid Earth, 2016_

## Referee Comment (RC1) · Anonymous Referee #1 · 1 Nov 2016

Dear colleague,

Please find below my comments on the manuscript entitled: "The Kenya Rift revisited: insights into lithospheric strength through data-driven 3D gravity and thermal modelling" by Judith Sippel et al. My comments are also attached in a pdf file.

General comments ———————— This is a clear and interesting paper which presents new results in the form of 3D gravity, thermal and rheological model for the Kenya rift area. However, a geological setting section is missing. In addition, it is unclear how the crustal structure is constrained. A discussion on the uncertainty of the crustal structure is missing.

[Figure]

1. The paper fits well the scope of SE. 2. The paper presents 3D density-gravity, thermal and rheological models of the Kenya rift area. To my knowledge this is new, this kind of modeling has never been done for this area before. 3. The authors draw several conclusions from their models which can help the understanding of the geology of the area. However I find some of their conclusions are weak because they are based on results which are not well constrained (crustal structure). 4. The scientific methods and assumptions are valid. 5. I find some of their results are weak. Notably, it is not really clear how the thickness and geometry of the different blocks in the upper and basal crustal layers are constrained though a lot of their interpretation is based on those results. 6. The descriptions of the models and methods are in general accurate. However, a description of the geometry of the model is missing (dimensions...) and the method for constraining the crustal structure (geometry and thickness of the different crustal blocks) is unclear. 7. Previous works are referenced well. 8. The title clearly reflects the content of the paper. 9. The abstract summarizes well the content of the paper. 10. The paper could be better organized. Notably a geological setting section is missing after the introduction. That would be useful for the reader to have a summary of the geology of the area and of the main geodynamic events. 11. The article is in general well written. The language is fluent and precise. 12. The mathematical formulae are correctly defined. 13. The tables are clear. 14. The reference list is complete and relevant. 15. Useful supplementary material is included in the form of four detailed appendixes.

Specific comments ——————— Part 1: The introduction is fine. However, the paragraphs (line 17, page 2 to line 14 page 3) describing the interactions between mantle dynamics and lithosphere is not clear. What kind of interactions are you talking about? (is it rifting? volcanism? doming?). It is not clear how your model can help to improve the understanding of these interactions. Please be more accurate. You consistently refer to western and eastern Kenya throughout the paper. What do they represent? Is western Kenya located west of the rift and eastern Kenya east of the rift? Please specify it at the beginning of the paper (in the introduction for example).

A section "Geological setting and/or history" is missing between the introduction and part 2. Such a section may be useful to readers who are not familiar with the geology of the Kenya rift area and the main geodynamic events (amalgamation, rifting episodes, plume emplacement. . .).

Part 2: At the beginning of part 2 some important information are missing such as the dimensions of the model and an accurate identification of the different density layers you are considering (for example, you should indicate that your modeled mantle has two layers, the first between Moho and 1000 km deep and the second between 100 km deep and 200 km deep. . .).. Line 21 (page 4) to line 17 (page 5): this part describing the basin formation would fit better in a "Geological setting" section. For the upper mantle density distribution (between Moho and 100 km deep): which data do you favor: KRISP or Achauer and Masson(2002)? Line 11 (page 9): why "starting" model? Do you test different density models for the mantle?

Part 2.3.2: I understand the density is computed for depth between 100 and 200km. Are those density depth-averaged? This is not clear...

Part 3.4: I understand that the density and thickness of the upper crustal layer blocks and basal crustal layer blocks are constrained in order to best-fit the observed gravity. However, the way it is done is not clear. Is it done manually? Have you used a specific method? Are the blocks consistent with the five tectono-thermal crustal domains? What is the uncertainty on the crustal structure? Please be more accurate. The crustal structure is important for your later discussion. What 's the reference density column for the gravity computation?

Part 3: The sentence: "the modeled thickness. . . . . . and the Moho geometry" (line 31-33 page 12) is not clear. Again, how are the different crustal blocks delineated? I understand those are constrained from KRISP data but how are they constrained away from the KRISP profiles?

Part 4.2: What is the error between the modeled and observed heat flow?

Part 5.1.1: The sentence "hence these local. . . . . . thickness maxima" (line 32 page 16 – line 2 page 17) is not clear.

Part 5.1.2: Conclusions are drawn from the upper and basal crustal layer density distribution. However, the data constrains are poor. So a discussion on the uncertainty of the crustal density distribution would be interesting.

Line 5 (page 18): you interpret mafic rocks below the Kenya rift though it is low Vp and low density. Could it be something else? What are the reasons to interpret this as mafic rock despite low Vp and density? Again, it is not clear how the thickness of the basal crustal layer is constrained away from the KRISP profiles though your interpretation is based a lot on this result.

Part 5.2.1: What are the depths of observed seismic peaks at the various points? This could be useful to include those peaks on the YSE profiles of fig. 9.

Line 9 page 24: it looks volcanism is offset towards western (and not eastern) boundary on the fig. 10a in the northern Kenya rift. . Line 11 -21 (page 23). The discussion on the plume-lithosphere interaction is not clear. That would be useful to indicate the location of plume impingement on a figure (figure 10 a for example). The link between the plume and strain localization is not clear. What is the link between the plume emplaced beneath a compositionally heterogeneous crust and the focusing of crustal thinning within the Southward tapering Arabian Nubian Shield? Line 16-18 (page 25): local areas of mass excess rather be related to ignored positive thermal anomalies?

Technical corrections ———————— Line 4 (page 5): "earliest extension" is written twice. Line 4 (page 5): "Paleo-Eocene" instead of Paloecene.

Figures: please add a title for each figure for clarity. Please add the names of the main structural features when it is relevant (Anza basin, Turkana Trough, etc. . .). Figure 5a: what is the grey square? Indicate on fig 10a the location of plume impingement.

Some comments on the references: ref. Allen and Allen 2009 or 2013? ref. Baker,

Mitchell and Williams 1988 is missing in the text. ref. Cacace et al. (2016): isn't it Cacace and Scheck-Wenderoth (2016)? ref. Melnick et al. (2015). in the reference list it is 2012. the ref. Morley et al. (1999) (line 27 page 2) is missing in the reference list. ref. Strecker et al. (1990) is missing. ref. Turcotte and Scubert (2014) is missing. ref. Goetze (1978) is missing. ref Onuonga et al. (1997) is missing in the reference list. ref. Halls et al. (1987) is missing in the reference list. ref. Burov(2011) is missing in the reference list. ref. Catuneaanu et al. (2005) is missing in the reference list. ref. Seton et al. (2012) is missing ion the reference list. ref. Fuchs et al. (2013) is missing in the reference list.

Please also note the supplement to this comment:
http://www.solid-earth-discuss.net/se-2016-139/se-2016-139-RC1-supplement.pdf
* * *

---

## Referee Comment (RC2) · G. Kimbell (Referee) · 2 Nov 2016

**GENERAL COMMENTS**

This is a nice example of an integrated geophysical study and is appropriate for publication after minor revision. The process of constructing a three-dimensional structural, thermal and rheological model for a large area and with limited constraints is, necessarily, subject to considerable uncertainty, but the authors show sound judgement in the methodology they develop and the interpretation they place on the results. The paper is well-written and the procedures are generally well-documented, although some possible improvements are identified below.

[Figure]

SPECIFIC COMMENTS

p5, line 4: 'Paleo-Eocene', is not common usage and should be spelled out. I'm not sure whether the Paleocene part of this is valid in the context of the references cited at the end of the sentence. The volcanism that pre-dates the rifting is dated at 45-39 Ma (middle Eocene) by Ebinger and Scholz (2012) and Morley et al. (1992) suggest the earliest rifting occurred in the Oligocene.

p7, line 2: Maybe identify the source of the gridded velocity model - (the GFZ Data Archive?)

p7, Eq. 1 (also Appendix B): It is justifiable to assume a linear relationship between velocity and density in the crystalline crust and it is correct to cite Birch (1961) in support of this. The chosen relationship provides densities that seem reasonable. However, the source of the constants quoted for the original Birch relationship is unclear as they do not match any of the solutions presented in the 1961 paper.

p8, line 7: The drho/dVp term in the Ravat (1999) equation for the mantle (Eq. 2) is much lower than the equivalent term in the $\sim$Birch equation for the crust (Eq. 1). Some justification for this should be included (proximity to the solidus with the former?)

p11, lines 10-14 (also p29, lines 3-4; footnote to Table 2): The way the Bouguer anomaly was calculated needs to be identified. If it was by assuming a uniform density of 2670 kg m-3 (the standard reduction with EIGEN-6C4) then the assumption that lateral density contrasts above datum can be ignored is not necessarily valid. It may well be that sensitivity analysis indicates that the inaccuracies involved are small compared with the scale of the anomalies under investigation, but that should be explained. Was the computation surface the sea-level datum?

p12: The method used for modifying the density structure of the upper and basal crustal layers and the top of the basal crustal layer should be described. Was this by manual adjustment or an automated procedure? There is inevitably a degree of

non-uniqueness in the way the adjustments are partitioned between the different variables and it might be better to describe the resulting densities as 'guided' rather than 'constrained' by the gravity data. Lateral changes in density are presumably better resolved than the absolute values, as there is a degree of trade-off between the latter and the way the background or reference model is defined (which also needs to be described).

p15, line 23: It would be advisable to present this initial reference to integrated strength in scientific notation as well as logarithmic notation (given that the former is more commonly used in other studies involving this parameter).

p16, lines 30-32: The maximum depth difference between LITHO1.0 and the present model appears to be underestimated, at least on the basis of visual inspection of the figures of Pasyanos et al. (2014). I recommend checking these figures.

p19-20: The model resolution is very coarse (50 x 50 km), raising the concern that this factor influences the details of the residual anomalies discussed on these pages. For example, the spatial relationship between a residual gravity low and the Nyanza Rift is discussed in some detail, but is actually only defined by a handful of model nodes. Not sure how this concern can best be addressed: maybe the authors should look at the more detailed gravity imaging of Mariita and Keller (2007) to see whether it helps with the analysis?

p21, line 14: Thermal modelling is difficult in this region, for the reasons the authors describe. In view of this it might be better to describe the steady-state conductive model they present as an 'appropriate general approximation' rather than a 'suitable approximation'. Was there a reason for using thermal gradients rather than heat flows for comparison with the observations?

p22, line 20: Is 'largely controlled' an overstatement? There is a thermal contribution as well, although its relative impact is difficult to judge.
p23, line 14: The authors have previously judged (p16) that it would be too speculative to implement lateral heterogeneities in mantle composition to assess the related influence on the gravity field. However, reference to the Ashwal and Burke (1989) hypothesis probably justifies at least a qualitative mention of its possible influence on the model, given that depleted mantle has a lower density than undepleted mantle for a given VS (Priestly and McKenzie, 2006).

General note for Section 5: There should be reference to the differences between present-day (modelled) conditions and those that applied at the time rifting was initiated.

References: I haven't checked these in any detail, but did notice that Goetze and Poirier (1978) should be Goetze (1978).

Figure 6: Have the authors considered also showing the calculated gravity field prior to model adjustment in this figure?

Figure 9: Is the sedimentary fill of the Kenya Rift properly represented in this figure? It only appears to be indicated by a slight deviation of the zero depth line.

TECHNICAL CORRECTIONS

p2, lines 31-33: suggest rewording this sentence

p3, line5: delete 'has' at end of line

p3, line 26: guided > guides

p5, line 4: 'earliest extension' repeated

p6, line 25 (and elsewhere): the authors of the model use the term 'LITHO1.0'

p8, line 9: 'the authors have' > 'those authors'?

p9, line 7: delete 'slightly'?

p9, line 30: '(2012) performed for a' > '(2012), which was performed on a'

p11, lines 2-3: suggest rewording this sentence

p13, Eq. 3: should a dot be used rather than an asterix in the heat equation?

p20, line 6: 'size' > 'extent'

p24, lines 3-6: suggest rewording this sentence

p25, line15: 'local mass defecits (positive gravity residuals >+30 mGal' > 'local positive gravity residuals (>+30 mGal'.

p25, line 20: delete 'one step'?

p26, line 5: not sure about 'strikingly'

p26, line 26: suggest deleting 'detailed'

Figure 1: maybe identify that II and III comprise the Mozambique belt

Figure 2 caption: 'and from a newly developed...' > 'with a ...' + add a reference to the source of the global sediment thickness information

Figure 2 caption: 'the topography, respectively bathymetry' > 'topography'?

Figure 4: the densities on the colour bars for (d) and (e) are in Mg/m3 rather than kg/m3

Figure 10: possible to use subscript for 10 in log10?

Table 5 caption: cross reference locations in Figure 10a?

Table C1: does XFe need explanation?

Geoff Kimbell

2 November 2016

---

## Editor Comment (EC1) · G. Peron-Pinvidic (Editor) · 9 Nov 2016

This manuscript fits very well the scope of SE contributions. It is clear, detailed and very well written. The study is original, new, well presented, very well referenced and should be of good interest to many of our colleagues. Two reviewers have listed various comments and recommendations (the 'RC' Referee Comments 1 and 2) that are very pertinent and should be highly useful to the authors to improve their contribution. As Editor, I have some additional recommendations:

1. I agree with the referees on the point that your initial model geometries should be better justified. Please add explanations, especially on how you defined your top-basement and the partition between your crustal layers. In that scope, please add

the KRISP profiles to your contribution (in the Appendix). These will help justifying your initial geometries and help the reader better understand the geological setting of the rift. You should also add information on the uncertainties regarding your input constraints and consequences for the your modelling approach (e.g. the geometries and thicknesses of your various layers). The density values in your input parameters are also crucial to your results. Please expand the explanations and justifications given on pages 5-7 regarding the density/velocity values : what are the uncertainties / error bars on the initial density/velocity values? how does that impact the model results and resolution? How the uncertainties of each input dataset have been handled in the final model? (e.g. values such as 6.325 km/s are extremely precise, is such precision realistic? what is the associated error bar?).

2. Please add a proper 'Geological Background / Tectonic Setting' section.

3. Your figures are very good, complete and clear, and very well referenced. I have only minor recommendations: - same comment as RC1: please add a title on each of the your figure so that the reader can easily and rapidly get what the maps are about. - please increase the font size of all your labels and texts (many of them will probably be difficult to read on the final version of the manuscript). - would it be possible to generate '3D perspective views' of your model? (e.g. with the topo/top-basement/Moho layers). That would help a lot the reader getting a good understanding of your geometries and of the rift configuration (to be placed for instance on the Figure 7?). - can you display the volcanoes on the Figure 8b as done for the Figure 6c? - add labels for your upper/basal crustal layers on Figure 9.

4. Your reference list is very complete. Many doi numbers are however missing. Coudl you carefully check and add the informtaion wherever needed?

Best regards, Gwenn Peron-Pinvidic Trondheim, November 9th 2016

---

## Author Comment (AC1) · 5 Dec 2016

**Authors' reply to RC1**

**Specific comments RC1**

Part 1:
**RC1-1:** The introduction is fine. However, the paragraphs (line 17, page 2 to line 14 page 3) describing the interactions between mantle dynamics and lithosphere is not clear. What kind of interactions are you talking about? (is it rifting? volcanism? doming?). It is not clear how your model can help to improve the understanding of these interactions. Please be more accurate.
We have rephrased and complemented the last part of the introduction of the revised manuscript to clarify this point:
"The models allow a straightforward spatial correlation between modelled strength heterogeneities and regional deformation structures of the rift, seismicity patterns and major locations of volcanic activity, which we briefly discuss in terms of causal relationships. To improve upon these aspects, future studies are planned, which will integrate the presented 3D models into numerical forward geodynamics experiments in order to test hypotheses on the entire Cenozoic deformation history of the study area. In any case, the model as it now stands already shows how far a compositionally heterogeneous crust has controlled lithospheric deformation and thus rift localisation and propagation processes."

**RC1-2:** You consistently refer to western and eastern Kenya throughout the paper. What do they represent? Is western Kenya located west of the rift and eastern Kenya east of the rift? Please specify it at the beginning of the paper (in the introduction for example).
We have provided a short paragraph at the end of the new chapter "2 Geological Setting" (see below) to describe *a priori* differences between western and eastern Kenya with respect to the topography, basement depth and the distribution of KRISP refraction seismic profiles. With respect to this point, we would like to add that, given the complexity of the rift system we could not draw any clear demarcation line to separate western from eastern Kenya in a way that might be consistent with all of the observations presented in the manuscript. In the discussion of the revised manuscript, we also refer to this distinction, while again its meaning depends on the property discussed and is best derived from the figures referred to.

**RC1-3:** A section "Geological setting and/or history" is missing between the introduction and part 2. Such a section may be useful to readers who are not familiar with the geology of the Kenya rift area and the main geodynamic events (amalgamation, rifting episodes, plume emplacement…).
We have added the section "2 Geological Setting" to the manuscript. Thereby, all information given in this section has been derived from existing sections, i.e. the sections "1 Introduction", "2.1 Constraints on the density configuration of the sedimentary and volcanic rocks" and "2.2 Constraints on the density configuration of the crystalline crust" of the original manuscript. This has helped not to unnecessarily increase the length of the manuscript. Please note the required adjustments also in these sections of the revised manuscript.
The new section "2 Geological Setting" of the revised manuscript reads as follows:
"The formation of the continental crust in East Africa dates back to the Neoproterozoic when the East African Orogeny (at ≈650-620 Ma) led to the amalgamation of numerous terranes to form central Gondwana (e.g., Fritz et al., 2013). This orogeny resulted from collisions of the Arabian Nubian Shield and its southward continuation, the Mozambique Belt (Holmes, 1951), with the Tanzania (Nyanzian) Craton to the west and the Azania microcontinent to the east. According to Fritz et al. (2013), five major tectonothermal domains of different Precambrian ages and lithologies are juxtaposed against each other in the study area (Fig. 1b). From W to E, these are

[revised manuscript text omitted]

To summarise, western Kenya has strongly been affected by Cenozoic mantle dynamics as becomes evident from the high topography (as a result of doming; Fig. 1a) and the narrow basement lows (graben structures; Fig. 2a). In contrast, eastern Kenya shows low topographies (Fig. 1a) and considerably broader and deeper basins (Fig. 2a) that largely trace back to Mesozoic times. For western Kenya, the KRISP seismic experiments provide distributed information on deep crustal structures, while for eastern Kenya such information is confined to the south-easternmost parts of the proposed microcontinent Azania (Fig. 1b)."

Part 2:

**RC1-4:** At the beginning of part 2 some important information are missing such as the dimensions of the model and an accurate identification of the different density layers you are considering (for example, you should indicate that your modeled mantle has two layers, the first between Moho and 100 km deep and the second between 100 km deep and 200 km deep…).

As the model dimensions and resolution have been chosen according to the results of the analysis of diverse datasets as described in the revised chapters 3.1-3.3, we provide more details on the model setup in chapter "3.4 3D gravity modelling" of the revised manuscript.

"We have used the above constraints on the structure and density of the sedimentary and volcanic cover (Fig. 2, Table 1, Table 2), the crystalline crust (Fig. 3, 4), and the mantle (Fig. 5) to set up a starting 3D density model. This model spans 850 km in E W direction and 1100 km in N S direction (black rectangle in Fig. 1a). To model discrete density bodies, the corresponding scattered information on delineating structural interfaces has been interpolated to regular grids of 50×50 km horizontal resolution. For example, the initial depth to the top of the Basal Crustal Layer has been obtained through interpolation (and extrapolation) of the corresponding KRISP refraction seismic information (Fig. 4a; Appendix B) to cover the entire continental crustal domain of the study area. In the same way, we have generated regular 50×50 km-grids for all first-order model layers, i.e. gridded tops for all sedimentary and volcanic units (Tab. 1, 2), the Upper Crustal Layer (Fig. 2a), and the mantle (Moho; Fig. 3a). Accordingly, the vertical resolution of the crustal parts of the generated 3D density model is variable as it is determined by the thicknesses of the different units. This applies also to the upper mantle domain between the Moho and 100 km, modelled by six units, each showing a constant density as derived from P-wave velocities (Fig. 5a). The S-wave derived density configuration of the lower mantle domain reaching from 100 to 200 km depth, on the other hand, is represented by point-wise density information, i.e. by the generated voxel grid with a regular spacing of 50 km horizontally and 20 km vertically.

The constant densities assigned to the modelled sedimentary and volcanic units are presented in Table 2, those of the shallowest mantle in Figure 5a. For the starting density model we have further chosen $\rho$=2750 kg m 3 for the Upper Crustal Layer (cf. Fig. 4d) and $\rho$=3000 kg m 3 for the Basal Crustal Layer (cf. Fig. 4e), while the oceanic crust has been assigned a value of $\rho$=2900 kg m $^3$. […]"

**RC1-5:** Line 21 (page 4) to line 17 (page 5): this part describing the basin formation would fit better in a "Geological setting" section.

We moved these paragraphs to the new chapter "2 Geological setting".

**RC1-6:** For the upper mantle density distribution (between Moho and 100 km deep): which data do you favor: KRISP or Achauer and Masson (2002)?

To clarify this, we have added the following sentence at the end of section 3.3.1 of the revised manuscript:

"Hence, we have chosen the overall larger density contrasts as indicated by the KRISP velocity profiles for the starting density model to be tested against the gravity field."

**RC1-7:** Line 11 (page 9): why "starting" model? Do you test different density models for the mantle?

Yes, we have tested different density configurations for the shallowest mantle (between the Moho and 100 km depth). An example is given in the second paragraph of section "5.5.1 Model sensitivity and robustness" of the original manuscript ("6.5.1" of the revised manuscript). There, we provide information on the calculated changes in the modelled gravity (50 mGal) corresponding to a decrease of the across-rift density contrast (from 25 kg m$^{-3}$ to 10 kg m$^{-3}$). The starting density model is consistent with the KRISP-velocity-derived densities. We have decided to present this mantle configuration also as the final model, since the larger KRISP mantle density contrasts (compared to the smaller contrasts of the tomographic model of Achauer and Masson, 2002) produce a gravity anomaly large enough to keep the crustal densities very close to those directly derived from KRISP velocities using Eq. (1) (Fig. 4d, e).

**RC1-8:** Part 2.3.2: I understand the density is computed for depth between 100 and 200km. Are those density depth-averaged? This is not clear...

In chapter 2.3.2 of the original manuscript, we state that "we have complemented the model of Adams et al. (2012) towards the N and E by the model of Fishwick (2010) and 3D interpolated the scattered point information to obtain a voxel grid of regular spacing of 50 km horizontally and 20 km vertically." For the 3D gravity modelling, we have made use of this voxel (3D) grid. To clarify this point, we have added a sentence at the end of the first paragraph of section "3.4 3D gravity modelling" of the revised manuscript that reads:

"The S-wave derived density configuration of the lower mantle domain reaching from 100 to 200 km depth, on the other hand, is represented by point-wise density information, i.e. by the generated voxel grid with a regular spacing of 50 km horizontally and 20 km vertically."

Part 3.4:

**RC1-9:** I understand that the density and thickness of the upper crustal layer blocks and basal crustal layer blocks are constrained in order to best-fit the observed gravity. However, the way it is done is not clear. Is it done manually? Have you used a specific method? Are the blocks consistent with the five tectono-thermal crustal domains? What is the uncertainty on the crustal structure? Please be more accurate. The crustal structure is important for your later discussion.

We agree with the reviewer in that the crustal configuration as derived from the gravity-modelling step is of primary relevance for the whole discussion on the lithospheric strength, its zonation and inferred ideas on the rifting process. Therefore, we decided to include a more detailed description explaining the 3D gravity modelling procedure, by revising the last three paragraphs of section "3.4 3D gravity modelling" of the revised manuscript as summarized in what follows.

"It is important to note that the main focus of our study is to assess the density configuration of the continental crystalline crust across the whole study area. Therefore, we have only modified the starting 3D density model by varying this particular structural domain. Indeed, we have found that a reasonable fit between calculated and observed gravity can be obtained when keeping the density configurations of the sedimentary and volcanic cover as well as the mantle domains fixed (Fig. 6; Section 4).

In order to reproduce the observed long-wavelength variations in the gravity field, we have systematically modified the crustal 3D density configuration in our model. For this purpose, we have followed a "step-wise approach" relying on the IGMAS+ software capabilities. First, we have modified the topology of the top Basal Crustal Layer at locations not constrained by the

KRISP refraction lines in an attempt to arrive at a better agreement between calculated and observed gravity anomalies. We have followed a procedure in which we have varied (i.e. increased or decreased) the thickness of the Basal Crustal Layer along the selected 2D working sections while keeping track of the calculated gravity response of the model. It is worth mentioning that with these first imposed changes to the starting density model, we did not alter the thickness of the whole crustal layer; instead, any imposed variation in the Basal Layer thickness was complemented by respective variations in the thickness of the Upper Crustal Layer.

In a second stage, we have checked and confirmed (see Section 4) that a further improvement of the model fit on first-order gravity anomalies can be obtained through the implementation of the trends observed in the P-wave velocity configurations of the Upper and Basal Crustal Layers (Fig. 4d, e). This integration of lateral variations of density within both crustal layers was systematically done while interactively quantifying the gravity response of the whole model to each modification step. In a final step, the Moho topology has been adjusted in order to improve the fit between modelled and observed gravity anomalies, though limited to an area of small lateral extent where no gravity-independent constraints were available (hatched area; Fig. 3a)."

As we agree with all reviewers that the manuscript requires a more detailed discussion of the uncertainties of the modelling results, we have added the following five paragraphs to the revised section "6.1.1 Model sensitivity and robustness":

"In our gravity modelling approach we consider one single degree of freedom, which is the density configuration of the crystalline crust. However, given the relationship between two differently dense crustal layers and the resulting gravity response, the solution to our problem requires to take into account an additional free parameter, which is the depth of the top of the Basal Crustal Layer outlining the thickness variations of the two layers. For this purpose, we present the map of the obtained average crustal density (Fig. 7c) together with the thicknesses and densities of the two crustal bodies (Fig. 7a, b). While the average crystalline crustal density (as derived from the density and thickness configurations of the two crustal units) may be regarded as the more appropriate interpretation of the observed gravity anomalies across wide parts of the study area, it under-interprets the structural constraints provided by the KRISP profiles in western Kenya.

In the final 3D model, as constrained via the conversion of P-wave velocities and by gravity modelling, lateral variations in the density configuration are more reliable than absolute density values. This is because of uncertainties inherent in the density structure considered as the starting model. The most important determined trend, however, in terms of density gradients between western and eastern Kenya (Fig. 7c), is consistently mapped by both an eastward increase in the thickness of the relatively denser Basal Crustal Layer and by the lateral density variations of the two crustal units.

The quality of the final modelling results rely on the quality of the input data used to build up the starting 3D density model. Uncertainties associated with each dataset are, however, partly unknown (such as for the basement depth; Beicip, 1987), different in type (similar to the data), and are also transferred in a different manner to the 3D model (via interpolation, velocity-density conversion etc.). All of this hampers a quantification of uncertainties. It is also worth noting that any gravity-guided manual adjustment to the density configuration is subject to the modeller's decision. Therefore, there is an inevitable degree of non-uniqueness in the way density variations are partitioned. Although we have carried out all modifications in a systematic way, the modelling approach does not permit any straightforward quantitative assessment of related uncertainties with respect to the final 3D density configuration.

The five tectono-thermal domains that are proposed to represent surface expressions of a complex juxtaposition of interlocked crustal units (Fritz et al., 2013) have not been used as input

for the 3D modelling. Since most of the study area is covered by Mesozoic-Cenozoic sediments and volcanics, the spatial distributions of these five domains (Fig. 1b) and their geometrical continuation towards greater crustal depths have only been interpreted from scattered outcrop observations (including fault geometries; Fritz et al., 2013). Our seismic velocity- and gravity-guided 3D density model for the first time provides the basis for a joint interpretation of deep geophysics and surface geological observations concerning the configuration of the crust across the entire study area (see section 6.1.2).

To summarise, we present a 3D density model that is not only consistent with the observed gravity field, but also cross-checked with a wide spectrum of gravity-independent criteria and observations. The strength of our modelling approach thus stems from an efficient integration and usage of a large variety of different datasets. Furthermore, as already discussed above, the obtained trends in crustal density heterogeneities would have remained of the same order even if the density configurations of the sediments and mantle would have been implemented differently from what was done in this study, though still within the respective data constraints."

**RC1-10:** What's the reference density column for the gravity computation?

The reference (background) density applied for the calculations of the gravity response of the 3D model is 3250 kg m$^{-3}$. IGMAS+ calculates gravity anomalies by considering densities of the 3D model as density anomalies with respect to this overall reference density. Hence, we have chosen the value for the reference density to represent an overall average density of the entire modelled volume.

In the revised manuscript, we have added this information to section 3.4 (third paragraph).

Part 3:

**RC1-11:** The sentence: "the modeled thickness… … and the Moho geometry" (line 31-33 page 12) is not clear. Again, how are the different crustal blocks delineated? I understand those are constrained from KRISP data but how are they constrained away from the KRISP profiles?

We have rephrased and complemented this sentence for the revised manuscript:

"The modelled thickness anomalies of the Basal Crustal Layer differ significantly in wavelength (<150 km) and spatial distribution from both its internal segmentation into four regional density domains (Fig. 7b) and the Moho geometry (Fig. 3a). This demonstrates that it was possible to differentiate between thickness and density characteristics of this layer since they conform to different components of the observed gravity field."

As we detail in the final paragraph of chapter 3.4 (revised manuscript), we have used the gravity signal to model crustal densities for the regions away from the KRISP seismic profiles. Nevertheless, the two different crustal blocks of the Upper Crustal Layer are not only consistent with the gravity anomaly pattern that indicates larger masses in the east (Fig. 6a), but also with an abrupt eastward increase of velocities along KRISP lines D, E, and F (Fig. 4d). After having split the Upper Crustal Layer, we have obtained an intermediate status of the residual gravity that indicated (i) mass excess along and around the northern parts of the rift, consistent with relatively lower velocities in the Basal Crustal Layer as revealed by KRISP (Fig. 4e) and (ii) mass deficits in the southeastern parts of the study area, consistent with higher velocities in the KRISP Basal Crustal Layer there (Fig. 4e). Based on this intermediate result, we have split the Basal Crustal Layer into four parts, two of them showing different densities than the one characterising the Basal Crustal Layer in the starting 3D density model (3000 kg m$^{-3}$).

We have refrained from describing all the intermediate modelling steps and results in the manuscript, as they would not improve the reliability of the final model, at least not more than the demonstrated consistency between the final model and the different observables, such as gravity (Fig. 6), seismic velocity (Fig. 4d, e) and geology (Azania's western margin; Fig. 7a).

Part 4.2:

**RC1-12:** What is the error between the modeled and observed heat flow?

The first-order result of calculating the 3D conductive thermal field is a temperature configuration. The first-order measurement in heat-flow assessments is temperature, too (Nyblade et al., 1990; Wheildon et al., 1994). For this reason, we compare model results and observations in terms of temperatures (geothermal gradient differences; Fig. 8c).
Comparing modelled and "measured" heat flow would mean to depart from the original modelling and measurement results, since in both cases heat flow is calculated based on Fourier's law, i.e. assuming a certain thermal conductivity as the coefficient to be multiplied with the (modelled or measured) thermal gradient.

Part 5.1.1:

**RC1-13:** The sentence "hence these local… … thickness maxima" (line 32 page 16 –line 2 page 17) is not clear.

Since the Moho depth presented by Woldetinsae (2005) is largely based on gravity modelling and due to its finer spatial resolution deviating locally from the global Moho model of Pasyanos et al. (2014), there is some uncertainty in fixing this important density discontinuity. The largest differences in Moho depth between the two models correspond with differences in gravity response that do not exceed 30 mGal. We have rephrased the following sentence for the revised manuscript:

"Hence, no matter which of these two models we had integrated, one of the main findings of this study would remain, namely that northeastern Kenya is regionally underlain by a lower crust of high density ($\rho$=3000 kg m$^{-3}$) with NW-SE oriented thickness maxima (Fig. 7b)."

Part 5.1.2:

**RC1-14:** Conclusions are drawn from the upper and basal crustal layer density distribution. However, the data constrains are poor. So a discussion on the uncertainty of the crustal density distribution would be interesting.

Here, we would like to refer to our comment to **RC1-11** about data constraints. Again, although the overall distribution of differently dense crustal domains is mainly gravity constrained, the existence of major contrasts in seismic velocity confirms these modelled density contrasts. Concerning the uncertainties of the modelling results (crustal density configuration), we would like to refer to our comment to **RC1-9**.

**RC1-15:** Line 5 (page 18): you interpret mafic rocks below the Kenya rift though it is low Vp and low density. Could it be something else? What are the reasons to interpret this as mafic rock despite low Vp and density?

With a density of 2920 kg m$^{-3}$ below the northern Kenya Rift, the Basal Crustal Layer does indicate mafic rocks there (given that gabbroic rocks typically show densities of 2900 kg m$^{-3}$, for instance). In the first paragraph of the sub-section "Basal Crustal Layer" in chapter 5.1.2 we refer to a number of other studies that have previously interpreted the seismically constrained Basal Crustal Layer as representing mafic intrusions related to the rifting process. At the end of this paragraph, we provide a possible explanation for the density/velocity to be low with respect to the values in the remaining study area (3000-3050 kg m$^{-3}$): they might be low "due to elevated mantle and crustal temperatures (cf. Fig. 5c, 8a) and related thermal expansion of the rocks (see also e.g. Maguire et al., 1994)".

**RC1-16:** Again, it is not clear how the thickness of the basal crustal layer is constrained away from the KRISP profiles though your interpretation is based a lot on this result.

Again, we would like to refer to our comment to **RC1-11** about data constraints.

Part 5.2.1:
**RC1-17:** What are the depths of observed seismic peaks at the various points? This could be useful to include those peaks on the YSE profiles of fig. 9.
For points E-H (Fig. 10a), Table 5 relates the "Modelled depth of the top of the brittle-ductile transition" (for the crust and mantle, respectively) to the observed "Depths of peak seismicity". For the locations of the YSE profiles, we do not have corresponding observations on seismicity. The locations of the YSE profiles have been chosen to span a structural profile across the Kenya Rift as illustrated by Figure 9.

**RC1-18:** Line 11-21 (page 23). The discussion on the plume-lithosphere interaction is not clear. That would be useful to indicate the location of plume impingement on a figure (figure 10 a for example).The link between the plume and strain localization is not clear. What is the link between the plume emplaced beneath a compositionally heterogeneous crust and the focusing of crustal thinning within the Southward tapering Arabian Nubian Shield?
The plume discussed here refers to a mantle thermal anomaly observed at the present day that reaches far beyond the limits of the modelled area. It is also known as the East African Superplume and was used by Koptev et al. (2015) to test the hypothesis that a mantle plume starting to rise beneath the Tanzania Craton may have been responsible for extensional tectonics and volcanism along the eastern branch of the East African Rift. The observed smaller-scale thermal anomaly underneath the Kenya Rift might be a derivative of this superplume and is exemplarily shown by the depth of the 1350°C-isotherm as derived from S-wave velocities which will give some indication on the plume impingement domain in the study area (Fig. 8a).

To clarify our discussion about plume-lithosphere interactions, we have rephrased and complemented the third paragraph of section 6.2.2 of the revised manuscript:
"The presented 3D model is the first to jointly integrate the present-day mantle thermal anomaly, crustal composition and related strength variations within the crust and lithospheric mantle. This opens the possibility for new hypotheses on plume-lithosphere interactions, i.e. on how dynamic mantle buoyancy forces contributed to tensional stresses in the lithosphere and how the latter responded. According to the 3D model, the plume-related lithospheric thinning would have been taking place beneath a compositionally and rheologically heterogeneous crust (Fig. 7; Table 4) – even though its structural configuration and, above all, its thermal state have certainly not been the same in the past. Crustal thinning obviously focussed within the southward tapering Arabian Nubian Shield (Fig. 1b) as the easternmost part of the rheologically weaker domain of western Kenya (Fig. 10a, b). At the same time, the configuration of eastern Kenya comprising Azania upper crust and remarkably thick, dense, and stiff lower crustal rocks (Fig. 7) might have formed a strong barrier against crustal deformation. Hence, strain localisation (induced by mantle dynamics and related tensional stresses) would have been facilitated by pre-existing contrasts in rheological properties between western and eastern Kenya."

**RC1-19:** Line 9 page 24: it looks volcanism is offset towards western (and not eastern) boundary on the fig. 10a in the northern Kenya rift.
This is true. We have rephrased the sentence to be more precise about the area we want to discuss:
"In the central Kenya Rift (i.e. just north of the Kenya Rift-Nyanza Trough junction; cf. Fig. 2a), this narrow zone of strongest crustal thinning and volcanism is locally offset from the rift centre towards the eastern boundary of the surface expression of the rift (Fig. 10b)."

**Technical corrections**
**RC1-20:** Line 4 (page 5): "earliest extension" is written twice.
Corrected.

**RC1-21:** Line 4 (page 5): "Paleo-Eocene" instead of Paloecene.
We have changed "Paleo-Eocene" to "Paleocene-Eocene".

**RC1-22:** Figures: please add a title for each figure for clarity.
We have rephrased the legend descriptions so that they function as titles.

**RC1-23:** Please add the names of the main structural features when it is relevant (Anza basin, Turkana Trough, etc…).
Done.

**RC1-24:** Figure 5a: what is the grey square?
The grey square indicated the XY-dimensions of the tomographic P-wave velocity model of Achauer and Masson (2002), which is indicated in the figure caption.

**RC1-25:** Indicate on fig 10a the location of plume impingement.
We have plotted the 125 km-contour of the 1350°C-isotherm in Fig. 10a to illustrate where the "plume" reaches shallow depth levels.

Some comments on the references:
**RC1-26:** ref. Allen and Allen 2009 or 2013?
In the text and references we refer to the latest version (Allen and Allen, 2013).

**RC1-27:** ref. Baker, Mitchell and Williams 1988 is missing in the text.
Deleted from the references list.

**RC1-28:** ref. Cacace et al. (2016): isn't it Cacace and Scheck-Wenderoth (2016)?
Corrected.

**RC1-29:** ref. Melnick et al. (2015). in the reference list it is 2012.
Corrected.

**RC1-30:** the ref. Morley et al. (1999) (line 27 page 2) is missing in the reference list.
In the text it should read as "Morley (1999)" as in the reference list. Corrected.

**RC1-31:** ref. Strecker et al. (1990) is missing.
Deleted from the reference list.

**RC1-32:** ref. Turcotte and Scubert (2014) is missing.
Added to the reference list.

**RC1-33:** ref. Goetze (1978) is missing.
Corrected to Goetze and Poirier (1978).

**RC1-34:** ref Onuonga et al. (1997) is missing in the reference list.
Added.

**RC1-35:** ref. Halls et al. (1987) is missing in the reference list.
Added.

**RC1-36:** ref. Burov (2011) is missing in the reference list.
Added.

**RC1-37:** ref. Catuneanu et al. (2005) is missing in the reference list.
Added.

**RC1-38:** ref. Seton et al. (2012) is missing ion the reference list.
Added.

**RC1-39:** ref. Fuchs et al. (2013) is missing in the reference list.
Added.

---

## Author Comment (AC2) · 5 Dec 2016

**Authors' reply to RC2**

SPECIFIC COMMENTS

**RC2-1:** p5, line 4: 'Paleo-Eocene', is not common usage and should be spelled out. I'm not sure whether the Paleocene part of this is valid in the context of the references cited at the end of the sentence. The volcanism that pre-dates the rifting is dated at 45-39 Ma (middle Eocene) by Ebinger and Scholz (2012) and Morley et al. (1992) suggest the earliest rifting occurred in the Oligocene.
We have changed "Paleo-Eocene" to "Paleocene-Eocene".

**RC2-2:** p7, line 2: Maybe identify the source of the gridded velocity model - (the GFZ Data Archive?)
There is no specific "source" to be referred to at this point. One of the authors, James Mechie, has archived the seismic data in the GFZ data archive since the 1990s. The models along the various KRISP profiles from which the gridded velocities are taken, are all published and referenced in Khan et al. (1999) and Khan et al. (1999) is referenced in the present paper.

**RC2-3:** p7, Eq. 1 (also Appendix B): It is justifiable to assume a linear relationship between velocity and density in the crystalline crust and it is correct to cite Birch (1961) in support of this. The chosen relationship provides densities that seem reasonable. However, the source of the constants quoted for the original Birch relationship is unclear as they do not match any of the solutions presented in the 1961 paper.
These constants have been derived from the measured properties as presented in Birch (1961, 1964).

**RC2-4:** p8, line 7: The drho/dVp term in the Ravat (1999) equation for the mantle (Eq. 2) is much lower than the equivalent term in the Birch equation for the crust (Eq. 1). Some justification for this should be included (proximity to the solidus with the former?)
Both equations represent empirical relationships between compressional wave velocities and densities of rocks. Birch's law and the empirical constants of it have been derived from general measurements on crustal and mantle rocks, while Ravat et al. developed their equation for the mantle rocks beneath the Kenya Rift. As the mantle rocks, much more so than the crustal rocks, beneath the Kenya Rift are probably much closer to the solidus than normal, drho/dVp can be expected to be significantly smaller than normal in the mantle here. Hence it is justifiable to use a different relationship.

**RC2-5:** p11, lines 10-14 (also p29, lines 3-4; footnote to Table 2): The way the Bouguer anomaly was calculated needs to be identified. If it was by assuming a uniform density of 2670 kg m-3 (the standard reduction with EIGEN-6C4) then the assumption that lateral density contrasts above datum can be ignored is not necessarily valid. It may well be that sensitivity analysis indicates that the inaccuracies involved are small compared with the scale of the anomalies under investigation, but that should be explained. Was the computation surface the sea-level datum?
We agree with the reviewer that we should be clearer about these issues, so we have implemented some modifications in the revised manuscript:

Added to the third paragraph of section "3.4 3D gravity modelling" of the revised manuscript:
"Details on the reduction of the original gravity data to obtain Free-air and Bouguer anomalies (e.g. assuming a constant density of 2670 kg m$^{-3}$ for the Bouguer plate) are presented on the website (http://icgem.gfz-potsdam.de/ICGEM/). To warrant comparability between the Bouguer anomalies and the calculated response of the 3D density model, all masses located above sea-level have been removed from the model (thus referring to the Bouguer plate above the geoid)."

Added to last paragraph of Appendix A:
"Further, although these density heterogeneities existing above sea-level have transferred inaccuracies to the Bouguer reduction of gravity data (which is based on a constant Bouguer plate density of 2670 kg m$^{-3}$), neglecting the volcanic edifices in our modelling is not an obstacle to uncovering deep crustal density anomalies due to the edifices' limited spatial extents compared to first-order gravity anomalies."

The footnote to Table 2 has been rephrased: FROM "#not considered for calculations of the Bouguer gravity since the volcanic edifices are located above sea level […]" in the original manuscript TO "#not considered for calculations of the gravity response of the 3D density model (see main text)".

**RC2-5:** p12: The method used for modifying the density structure of the upper and basal crustal layers and the top of the basal crustal layer should be described. Was this by manual adjustment or an automated procedure?
We would like to refer to the first part of our comments to **RC1-9**.

**RC2-6:** There is inevitably a degree of non-uniqueness in the way the adjustments are partitioned between the different variables and it might be better to describe the resulting densities as 'guided' rather than 'constrained' by the gravity data. Lateral changes in density are presumably better resolved than the absolute values, as there is a degree of trade-off between the latter and the way the background or reference model is defined (which also needs to be described).
We would like to refer to the second part of our comments to **RC1-9**.
The reference model is described in section 3.4 of the revised manuscript.

**RC2-7:** p15, line 23: It would be advisable to present this initial reference to integrated strength in scientific notation as well as logarithmic notation (given that the former is more commonly used in other studies involving this parameter).
Added.

**RC2-8:** p16, lines 30-32: The maximum depth difference between LITHO1.0 and the present model appears to be underestimated, at least on the basis of visual inspection of the figures of Pasyanos et al. (2014). I recommend checking these figures.
We have followed this advice and checked Figure 8b of Pasyanos et al. (2014) showing global crustal thickness of the final LITHO1.0 model. Our comparison of the models, however, was based on the downloaded grid of Moho depth published as a "supplement" to Pasyanos et al. (2014). It seems that Figure 8b does not reflect the full resolution of the grid (it does not even show the Kenya Rift anomaly) and so there are major differences with respect to the grid. As an example, while the figure suggests that the Moho in NE Kenya is widely located at depths shallower than 35 km, only one of 11 grid points for this domain show depths of <35 km, while six points even exceed 39 km (up to 45km). For this reason, the figure suggests that the Moho of LITHO1.0 is shallower than the presented model in this area, while the grid points of the global model mostly plot below the Moho of our model.

**RC2-9:** p19-20: The model resolution is very coarse (50 x 50 km), raising the concern that this factor influences the details of the residual anomalies discussed on these pages. For example, the spatial relationship between a residual gravity low and the Nyanza Rift is discussed in some detail, but is actually only defined by a handful of model nodes. Not sure how this concern can best be addressed: maybe the authors should look at the more detailed gravity imaging of Mariita and Keller (2007) to see whether it helps with the analysis?
The horizontal resolution of the density model has been chosen according to (i) the distribution of available data (most importantly KRISP and the gravity anomalies) and (ii) the dimensions of the type of density domains to be "mapped" (regional basement domains). Increasing the resolution would

have meant over-interpreting data (e.g. velocity anomalies) for wide parts of the study area. Because of the scatter of constraining data, we have decided not to fit gravity anomalies remaining under approximately 250 km (half-wavelength). We find that the modelled residual gravity already brings along much significance as it indicates areas of (probable) mass deficits and excesses. Any further investigations of those would require additional (geophysical) data and the model points to the regions where these data should be acquired.

Concerning the Nyanza Rift, this area of pronounced mass excess in the final density model is defined by 24 model nodes forming a consistent structure. It can well be seen in the observed gravity. However, any earlier investigations of this wider central Kenya rift area have missed covering the area indicated by the residual gravity anomaly. KRISP lines D and G run past this area, either further north or further south, respectively. Also the gravity analysis study of Mariita and Keller (2007) including valuable density profiles miss this part of the study area, probably since these studies integrated the knowledge gained by the KRISP campaigns.

**RC2-10:** p21, line 14: Thermal modelling is difficult in this region, for the reasons the authors describe. In view of this it might be better to describe the steady-state conductive model they present as an 'appropriate general approximation' rather than a 'suitable approximation'.
We have rephrased this according to the reviewer's suggestion.

**RC2-11:** Was there a reason for using thermal gradients rather than heat flows for comparison with the observations?
As copied from our reply to RC1-12: The first-order result of calculating the 3D conductive thermal field is a temperature configuration. The first-order measurement in heat-flow assessments is temperature, too (Nyblade et al., 1990; Wheildon et al., 1994). For this reason, we compare model results and observations in terms of temperatures (geothermal gradient differences; Fig. 8c). Comparing modelled and "measured" heat flow would mean to depart from the original modelling and measurement results, since in both cases heat flow is calculated based on Fourier's law, i.e. assuming a certain thermal conductivity as the coefficient to be multiplied with the (modelled or measured) thermal gradient.

**RC2-12:** p22, line 20: Is 'largely controlled' an overstatement? There is a thermal contribution as well, although its relative impact is difficult to judge.
Rephrased to "strongly controlled".

**RC2-13:** p23, line 14: The authors have previously judged (p16) that it would be too speculative to implement lateral heterogeneities in mantle composition to assess the related influence on the gravity field. However, reference to the Ashwal and Burke (1989) hypothesis probably justifies at least a qualitative mention of its possible influence on the model, given that depleted mantle has a lower density than undepleted mantle for a given VS (Priestly and McKenzie, 2006).
Ashwal and Burke (1989) present a model scenario that might explain why volcanism seems to be restricted to the off-cratonic areas of Africa. They relate volcanic activity to the presence of fertile mantle as opposed to cratonic areas showing depleted mantle compositions. We do not discuss this point further for two reasons: (i) For such a discussion, one would need a larger model that includes cratonic and off-cratonic domains in equal measure. Our model only integrates a relatively small portion of the Tanzania Craton – not enough for such a discussion as we think. (ii) We prefer to directly integrate smaller-wavelength mantle heterogeneities as indicated by the P- and S-wave velocity data. Beside their strong temperature-dependence, these variations in mechanical behaviour are also known to be compositionally-driven, even if less strongly (Priestley and McKenzie, 2006).

**RC2-14:** General note for Section 5: There should be reference to the differences between

present-day (modelled) conditions and those that applied at the time rifting was initiated.

To emphasize this point, we have included a sentence in the fourth paragraph of chapter 6.2.2 of the revised manuscript in the following way:

"According to our present-day the 3D model, the plume-related lithospheric thinning would have been taking place beneath a compositionally and rheologically heterogeneous crust (Fig. 7; Table 4) – even though its structural configuration and, above all, its thermal state have certainly not been the same in the past"

**RC2-15:** References: I haven't checked these in any detail, but did notice that Goetze and Poirier (1978) should be Goetze (1978).

Corrected. See also RC1-33.

**RC2-16:** Figure 6: Have the authors considered also showing the calculated gravity field prior to model adjustment in this figure?

As this would be only one of a large number of figures that would represent preliminary stages/results in the modelling procedure, we refrain from showing any of them (for the sake of a concise paper). Possible figures would be, e.g. gravity response of a model with a homogeneous crust, depth distribution of the top of the Basal Crustal Layer after (purely mathematical) interpolation/extrapolation, gravity response of a model with a two-layered crust, etc.

**RC2-17:** Figure 9: Is the sedimentary fill of the Kenya Rift properly represented in this figure? It only appears to be indicated by a slight deviation of the zero depth line.

Indeed, the thicknesses of sediments in western Kenya and along this line do not exceed 3000 m, while the topography is locally >3000 m.a.s.l. Thus, the figure correctly represents the model.

TECHNICAL CORRECTIONS

**RC2-18:** p2, lines 31-33: suggest rewording this sentence

Rephrased to: "However, detailed regional-scale assessments on the interaction between mantle dynamics and the overlying compositionally heterogeneous lithosphere in order to explain rift localisation and the mechanical predisposition of fault propagation have not been attempted yet for the region."

**RC2-19:** p3, line5: delete 'has' at end of line

Outdated as this part is rephrased as part of the new chapter "2 Geological Setting".

**RC2-20:** p3, line 26: guided > guides

Done.

**RC2-21:** p5, line 4: 'earliest extension' repeated

Corrected.

**RC2-22:** p6, line 25 (and elsewhere): the authors of the model use the term 'LITHO1.0'

Corrected.

**RC2-23:** p8, line 9: 'the authors have' > 'those authors'?

Corrected.

**RC2-24:** p9, line 7: delete 'slightly'?

Done.

**RC2-25:** p9, line 30: '(2012) performed for a' > '(2012), which was performed on a'

Corrected.

**RC2-26:** p11, lines 2-3: suggest rewording this sentence
Rephrased.

**RC2-27:** p13, Eq. 3: should a dot be used rather than an asterix in the heat equation?
Corrected.

**RC2-28:** p20, line 6: 'size' > 'extent'
Rephrased.

**RC2-29:** p24, lines 3-6: suggest rewording this sentence
Rephrased.

**RC2-30:** p25, line15: 'local mass defecits (positive gravity residuals >+30 mGal' > 'local positive gravity residuals (>+30 mGal'.
Rephrased.

**RC2-31:** p25, line 20: delete 'one step'?
Deleted.

**RC2-32:** p26, line 5: not sure about 'strikingly'
Deleted.

**RC2-33:** p26, line 26: suggest deleting 'detailed'
Deleted.

**RC2-34:** Figure 1: maybe identify that II and III comprise the Mozambique belt
Done.

**RC2-35:** Figure 2 caption: 'and from a newly developed...' > 'with a ...' + add a reference to the source of the global sediment thickness information
Rephrased and added.

**RC2-36:** Figure 2 caption: 'the topography, respectively bathymetry' > 'topography'?
Done.

**RC2-37:** Figure 4: the densities on the colour bars for (d) and (e) are in Mg/m3 rather than kg/m3
Corrected.

**RC2-38:** Figure 10: possible to use subscript for 10 in log10?
Done.

**RC2-39:** Table 5 caption: cross reference locations in Figure 10a?
Done.

**RC2-40:** Table C1: does XFe need explanation?
Explanation added.

---

## Author Comment (AC3) · 5 Dec 2016

**Authors' reply to EC1**

**EC1-1:** I agree with the referees on the point that your initial model geometries should be better justified. Please add explanations, especially on how you defined your top-basement and the partition between your crustal layers.

The first paragraph of the revised chapter "3.1 Constraints on the density configuration of the sedimentary and volcanic rocks" (chapter 2.1 of the original manuscript) details which data have been used to define the top of the crystalline basement. We added:

"The scattered data from inside and outside Kenya have been jointly interpolated to obtain a continuous regular grid (of originally 5×5 km horizontal resolution) of basement depths covering the entire study area (Fig. 2a)."

In addition, the caption of Figure 2 says: "all interpolations in this study have been performed using the Convergent Interpolation algorithm implemented in Petrel (©Schlumberger)".

More detailed information on the way how crustal densities have been found (domains partitioned) by manually adjusting the geometries / densities of model units to fit the observed gravity field has been provided in chapter "3.4 3D gravity modelling" of the revised manuscript. Refer also to our reply to **RC1-9**.

**EC1-2:** In that scope, please add the KRISP profiles to your contribution (in the Appendix). These will help justifying your initial geometries and help the reader better understand the geological setting of the rift.

We have added figures of the KRISP profiles ABC, E, F, and G as Appendix B to the revised manuscript. We also plan to provide the original data in a digital form (as ASCII files) in the Supplements of the revised manuscript.

**EC1-3:** You should also add information on the uncertainties regarding your input constraints and consequences for your modelling approach (e.g. the geometries and thicknesses of your various layers).

Please refer to our comments to **RC1-9**.

**EC1-4:** The density values in your input parameters are also crucial to your results. Please expand the explanations and justifications given on pages 5-7 regarding the density/velocity values: what are the uncertainties / error bars on the initial density/velocity values? How does that impact the model results and resolution? How the uncertainties of each input dataset have been handled in the final model?

Again, please refer to our comments to **RC1-9**.

As the KRISP profiles and related velocity-depth distributions have been published in various articles that also inform about the technical details and uncertainties in the data acquisition and processing and interpretation, we refer to these articles. A review and overall interpretation of the velocity profiles used in this study has been provided by Khan et al. (1999). Further sources of reading on the KRISP campaigns can be found there.

For the conversion of seismic velocity to density, a linear function equivalent to Birch's law has been used (Eq. 1). The obtained point-wise information on crustal densities already indicates major lateral differences within the Upper and the Basal Crustal Layers (Fig. d, e). For the starting density model, these lateral variations have been neglected, assigning a constant density of 2750 kg/m³ to the Upper Crustal Layer and 3000 kg/m³ to the Basal Crustal Layer, as described in section 3.4 (2.4) of the

revised (original) manuscript. This "starting" contrast may have some influence on the manual density/gravity adjustment process as described in the last three paragraphs of section "3.4 3D gravity modelling" of the revised manuscript copied to our reply to **RC1-9**.

**EC1-5:** (e.g. values such as 6.325 km/s are extremely precise, is such precision realistic? what is the associated error bar?).
We have estimated average velocities in eastern and western Kenya, i.e. east and west of the dashed line in Figure 4d, by analysing the point-wise available velocity values and inspecting their frequency distributions (medians in the histograms). In this way we derived "representative mean values" for the two domains. Although the accuracy of this method leads to the values given, we agree that this precision is not necessarily corresponding to the data as they have been processed (e.g. vertically interpolated) beforehand. For this reason, we have changed the values to $v_{p,c} \approx 6.33$ km s$^{-1}$ and $v_{p,c} \approx 6.43$ km s$^{-1}$, respectively.

**EC1-6:** Please add a proper 'Geological Background / Tectonic Setting' section.
Done.

**EC1-7:** Your figures are very good, complete and clear, and very well referenced. I have only minor recommendations: - same comment as RC1: please add a title on each of your figure so that the reader can easily and rapidly get what the maps are about.
Done.

**EC1-8:** please increase the font size of all your labels and texts (many of them will probably be difficult to read on the final version of the manuscript).
Done.

**EC1-9:** Would it be possible to generate '3D perspective views' of your model? (e.g. with the topo/top-basement/Moho layers). That would help a lot the reader getting a good understanding of your geometries and of the rift configuration (to be placed for instance on the Figure 7?).
Done.

**EC1-10:** can you display the volcanoes on the Figure 8b as done for the Figure 6c? –
Done

**EC1-11:** add labels for your upper/basal crustal layers on Figure 9.
Done

**EC1-12:** Your reference list is very complete. Many doi numbers are however missing. Could you carefully check and add the information wherever needed?
Done